# PKM2 aggregation drives metabolism reprograming during aging process

Juntao Bie[1,2,14], Ridong Li[3,14], Yutong Li[1], Chen Song[1], Zhaoming Chen [4], Tianzhuo Zhang[5], Zhiheng Tang[6], Li Su[7], Liangyi Zhu[8], Jiaxin Wang[9], You Wan [9,10], Jun Chen [11,15] ✉, Xiaoyun Liu [6,12,15] ✉, Tingting Li [4,10,15] ✉ & Jianyuan Luo [1,2,13,15] ✉

While protein aggregation's association with aging and age-related diseases is well-established, the specific proteins involved and whether dissolving them could alleviate aging remain unclear. Our research addresses this gap by uncovering the role of PKM2 aggregates in aging. We find that PKM2 forms aggregates in senescent cells and organs from aged mice, impairing its enzymatic activity and glycolytic flux, thereby driving cells into senescence. Through a rigorous two-step small molecule library screening, we identify two compounds, K35 and its analog K27, capable of dissolving PKM2 aggregates and alleviating senescence. Further experiments show that treatment with K35 and K27 not only alleviate aging-associated signatures but also extend the lifespan of naturally and prematurely aged mice. These findings provide compelling evidence for the involvement of PKM2 aggregates in inducing cellular senescence and aging phenotypes, and suggest that targeting these aggregates could be a promising strategy for anti-aging drug discovery.

Aging is a progressive process characterized by the systemic deterioration of organs and tissues, often culminating in chronic diseases such as diabetes, cancer, cardiovascular disorders, and neurodegenerative diseases[1,2]. In recent years, the disruption of proteostasis has emerged as a well-recognized hallmark of aging[3–7]. It is principally guarded by the chaperone-mediated folding system and degradation pathways involving lysosomes or proteasome[7–9]. Dysfunction in either of these systems can precipitate the accumulation of aberrant protein aggregates within cells, thereby contributing to the onset and progression of aging-related pathologies such as Alzheimer's disease,

[1]Department of Medical Genetics, Center for Medical Genetics, Peking University Health Science Center, Beijing 100191, China. [2]Medical Innovation Center (Taizhou) of Peking University, Taizhou 225316, China. [3]Institute of Systems Biomedicine, Beijing Key Laboratory of Tumor Systems Biology, School of Basic Medical Sciences, Peking University Health Science Center, Beijing 100191, China. [4]Department of Biochemistry and Biophysics, School of Basic Medical Sciences, Peking University Health Science Center, Beijing 100191, China. [5]Department of Anesthesiology, Key laboratory of Carcinogenesis and Translational Research (Ministry of Education/Beijing), Peking University Cancer Hospital & Institute, Beijing 100142, China. [6]Department of Microbiology and Infectious Disease Center, School of Basic Medical Sciences, Peking University Health Science Center, Beijing 100191, China. [7]Peking university medical and health analysis center, Beijing 100191, China. [8]Department of Pathology, School of Basic Medical Sciences, Peking University Third Hospital, Peking University Health Science Center, Beijing 100191, China. [9]Neuroscience Research Institute, Department of Neurobiology, School of Basic Medical Sciences, Peking University, Beijing 100191, China. [10]Key Laboratory for Neuroscience, Ministry of Education/National Health Commission of China, Peking University, Beijing 100191, China. [11]Peking University Research Center on Aging, Beijing Key Laboratory of Protein Posttranslational Modifications and Cell Function, Department of Biochemistry and Biophysics, School of Basic Medical Science, Peking University, Beijing 100191, China. [12]NHC Key Laboratory of Medical Immunology, Peking University, Beijing 100191, China. [13]Beijing Key Laboratory of Protein Posttranslational Modifications and Cell Function, Department of Biochemistry and Biophysics, School of Basic Medical Sciences, Peking University Health Science Center, Beijing 100191, China. [14]These authors contributed equally: Juntao Bie, Ridong Li. [15]These authors jointly supervised this work: Jianyuan Luo, Tingting Li, Xiaoyun Liu, Jun Chen. ✉e-mail: cjbiochem@bjmu.edu.cn; xiaoyun.liu@bjmu.edu.cn; litt@hsc.pku.edu.cn; luojianyuan@bjmu.edu.cn

Parkinson's disease, and cataracts[4,6]. Specific protein aggregates implicated in aging-related pathologies have been identified, including Tau, amyloid-β (Aβ), and α-synuclein (α-syn). Tau aggregation and Aβ plaques are widely acknowledged hallmarks of Alzheimer's disease[10], whereas α-synuclein aggregation is a prevalent feature in Parkinson's disease and dementia with Lewy bodies (DLB)[11]. Furthermore, age-related cataracts are closely associated with the aggregation of γ-crystallins in the lens[12]. However, in the context of organismal aging, while protein aggregation has been established as a common phenomenon in the aging process[13], specific protein aggregates contributing to aging have not been extensively reported in mammals. Notably, a singular study demonstrated that the aggregation of Whi3, an RNA-binding protein, induced aging in yeast cells[14].

To date, there have been no therapies targeting specific protein aggregates for anti-aging purposes. However, researchers have focused on discovering agents to dissolve aggregates implicated in aging-related pathologies. In the late 1990s, phenothiazines were found to inhibit tau aggregation by blocking tau-tau interactions[15]. Over the ensuing decades, numerous compounds have been identified for their ability to dissolve aggregates of tau or Aβ[16]. Peptides containing amino acid residues from the α-synuclein aggregation interface have been designed to dissolve α-synuclein aggregates[17]. Additionally, several small molecules, including curcumin, have been reported to inhibit α-synuclein aggregation[18–20]. Many of these compounds that impair α-synuclein aggregation also inhibit the formation of Aβ and tau filaments[18]. Among these aggregate inhibitors, epigallocatechin gallate (EGCG) has garnered significant attention for its ability to dissolve aggregates of α-synuclein, tau, and Aβ[21,22]. In the case of cataracts, lanosterol, an amphiphatic molecule enriched in the lens, has been shown to significantly reduce protein aggregates and restore eye function[23]. Similarly, targeting the dissolution of protein aggregates that accelerate aging may hold promise for the development of anti-aging agents.

In this study, we conducted an analysis of lysosomal proteomics from young and senescent cells, leading us to uncover the role of Pyruvate Kinase M2 (PKM2) aggregates in the aging process. In senescent cells, PKM2 tends to aggregate along with other glycolytic enzymes, such as Glyceraldehyde-3-phosphate dehydrogenase (GAPDH), α-Enolase (ENO1), and others. Furthermore, we found that PKM2 aggregation accompany with impairment of PKM2 enzymatic activity and glycolytic flux in senescent cells, exacerbating senescent phenotypes. To identify compounds capable of dissolving PKM2 aggregates and alleviating senescence, we conducted a series of screenings. K35 and its analog K27 were identified as compounds capable of inhibiting the formation of PKM2 aggregates. Treatment with K35 or K27 restored PKM2 enzymatic activity and glycolytic flux. Further studies demonstrated that K35 or K27 not only alleviated cellular senescence but also extended the lifespan of mice.

## Results
### Lysosomes in senescent cells contain increased glycolytic enzymes than in young cells
The impairment of lysosomal function in senescent cells[24] suggests that protein aggregates associated with senescence may accumulate in lysosomes without being digested. To identify potential aggregated proteins related to cellular senescence or aging, lysosomes were isolated from young and senescent cells and subjected to proteomic analysis using mass spectrometry. The differential expression of proteins was validated by Western blot and immunofluorescent imaging (Fig. 1a). By stably overexpressing TMEM192-3HA, a protein mainly distributed on the lysosomal membrane, lysosomes could be immunoprecipitated using HA magnetic beads[25]. The efficiency and purity of lysosomal isolation were examined by immunoblotting of corresponding organelle markers (Supplementary Fig. 1a). Low doses of etoposide were applied to induce senescence in HEK 293T cells, which had TMEM192-3HA stably overexpressed beforehand. The

upregulation of p21 protein levels (Supplementary Fig. 1b), *p21* and senescence-associated secretory phenotype (SASP) mRNA levels (Supplementary Fig. 1c), and increased staining of senescence-associated β-galactosidase (SA-β-gal) (Supplementary Fig. 1d) demonstrated successful establishment of etoposide-induced senescence in HEK 293T (TMEM192-3HA) cells. Lysosomal proteins from young and etoposide-induced senescent cells were extracted for mass spectrometry-based protein identification (Supplementary Fig. 1e). Analysis of the mass spectrometry data revealed that most glycolytic proteins were upregulated in lysosomes of senescent cells (Fig. 1b, c), including PKM2 rather than PKM1 (Supplementary Fig. 1f). Western blot analysis revealed that these glycolytic enzymes gradually enriched in lysosomes during the etoposide-induced senescence process (Fig. 1d). Additionally, immunofluorescent imaging showed that both PKM2 (Fig. 1e, f) and Glucose-6-phosphate isomerase (GPI) (Supplementary Fig. 1g, h) gradually entered lysosomes following etoposide treatment. Overall, most glycolytic enzymes gradually accumulated in lysosomes during etoposide-induced senescence.

### PKM2 aggregates accumulate in senescent cells and tissues of aged mice
Of the glycolytic enzymes that were enriched in lysosomes under etoposide treatment, PKM2 has been reported to be related with cellular senescence[26]. Besides, we found that PKM2 knockdown increased the staining of SA-β-gal (Supplementary Fig. 1i, j), and upregulated p21 protein level (Supplementary Fig. 1k), which indicated that PKM2 deficiency promoted cellular senescence. Consequently, PKM2 might be the potential protein we were looking for. Next, we thoroughly investigate the aggregating possibility of PKM2 in the aging process. Indeed, we observed that PKM2 tended to aggregate in etoposide-induced senescent cells (Fig. 2a, Supplementary Fig. 2a–c) as well as other genotoxic agents-induced senescent cells (Supplementary Fig. 2d) or ROS-induced senescent cells (Supplementary Fig. 2e). Moreover, PKM2 aggregates also increased during the progression of replicative senescence (Fig. 2b) or oncogene-induced senescence (Supplementary Fig. 2f) in human fibroblasts. Also, we found that PKM1 could form protein aggregates (Supplementary Fig. 2g). By high-content imaging, we also found the stably overexpressed sfcherry-PKM2 gradually aggregated in HeLa cells following exposure to etoposide (2 μM), which meant the formation of PKM2 aggregates was dynamic (Supplementary Fig. 2h, i). In addition to cellular senescence, PKM2 exhibited more aggregation in the liver and brain tissue of aged mice (Fig. 2c, Supplementary Fig. 3a). Outside of the cellular system, PKM2 was able to aggregate in vitro (Supplementary Fig. 3b) which was regulated by protein and salt concentration (Fig. 2d). As the confocal images (Fig. 1e) showed that not all PKM2 aggregates merged with lysosomes, it was unlikely that the appearance of PKM2 aggregation was due to its accumulation in lysosomes. To exclude the possibility that PKM2 aggregates might be mistaken because of its translocation to mitochondria, PKM2 and citrate synthase (CS) were stained together and then subjected to immunofluorescent imaging. The lack of merging of PKM2 and CS indicated that PKM2 aggregates were not related to mitochondria (Fig. 2e). Similarly, we found that PKM2 aggregates differed from stress granules (Supplementary Fig. 3c) by staining PKM2 together with stress granule marker, G3BP1. PKM2 protein is usually divided into N, A1, B, A2 and C domains (Fig. 2f). Different PKM2 truncated variants deleted with one of these domains were constructed and subjected to observing the possibility of PKM2 aggregation in vivo and in vitro. PKM2 with A1 or A2 or C domain deletion tended to form abnormal plaque in vivo (Fig. 2g, Supplementary Fig. 3d) and in vitro (Fig. 2h, Supplementary Fig. 3e) which revealed that these three domains were vital for the propensity of PKM2 aggregating. As for the influence on cellular senescence of the truncated PKM2, transfection of PKM2 constructs with A1 or A2 or C domain deletion resulted in upregulation of SASP mRNA levels

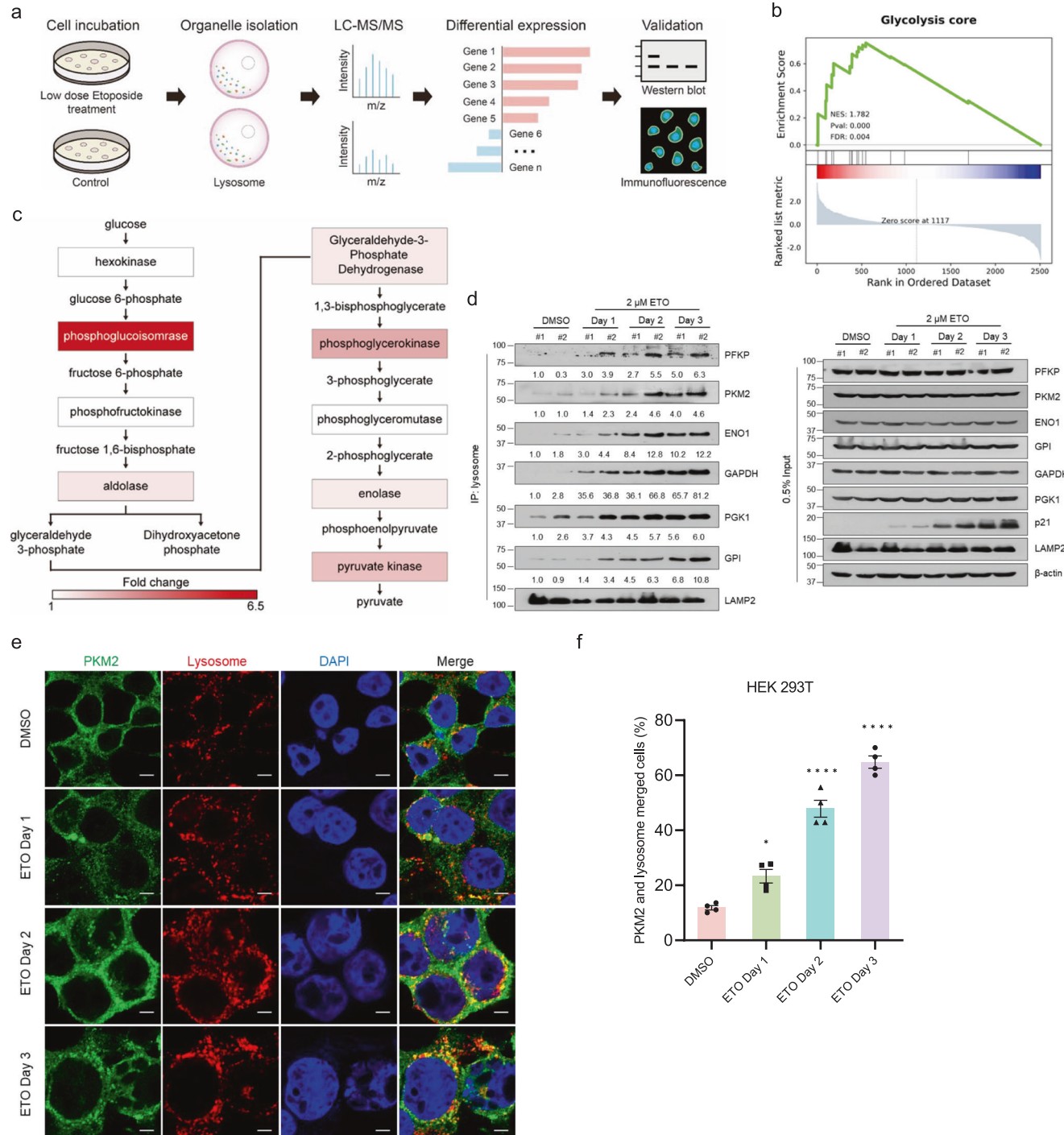

**Fig. 1 | Lysosomes in senescent cells contain increased glycolytic enzymes than in young cells. a** schematic diagram shows the process of isolating lysosomes from young and etoposide-induced senescent cells and the subsequent identification and verification. **b** GSEA plot of core components in glycolysis pathway according to the fold change scores in (**a**). **c** The Glycolysis pathway in which the increased glycolytic enzymes in lysosomes of senescent cells were marked in different levels of red. **d** Lysosome immunoprecipitation and subsequent immunoblotting analysis in HEK 293T (TMEM192-3HA) cells treated with 2 μM etoposide (ETO) for indicated days. **e**, **f** Representative images (**e**) and quantification (**f**) of PKM2 and lysosomes (indicated by HA staining) in HEK 293T (TMEM192-3HA) cells treated with 2 μM etoposide (ETO) for indicated days. Scale bar, 5 μm. *n* = 4 (four randomly captured images), one-way ANOVA was used, *P* = 0.0107 (DMSO vs. ETO Day 1). **P* < 0.05, *****P* < 0.0001. Error bars represent SEM. Source data are provided as a Source Data file.

(Supplementary Fig. 3f), upregulated p21 protein level (Supplementary Fig. 3g) and increased staining of SA-β-gal (Supplementary Fig. 3h, i), which are common features of senescent cells. To directly evaluate the effects of PKM2 aggregates on senescence or aging in vivo, we over-expressed wild-type Pkm2 and aggregate-prone Pkm2 mutant (A2 domain deletion) in mice via AAV9 system. The body weight of these mice was recorded every week (Supplementary Fig. 3j). After 14 weeks, the mice were subjected to physical test and then sacrificed for an evaluation of senescent markers. The results showed that the mice transduced with aggregate-prone Pkm2 mutant (A2 domain deletion) exhibited weaker physical fitness (Fig. 2i), upregulated level of SASP factors (Fig. 2j) and p21, p16 mRNA (Supplementary Fig. 3k), which

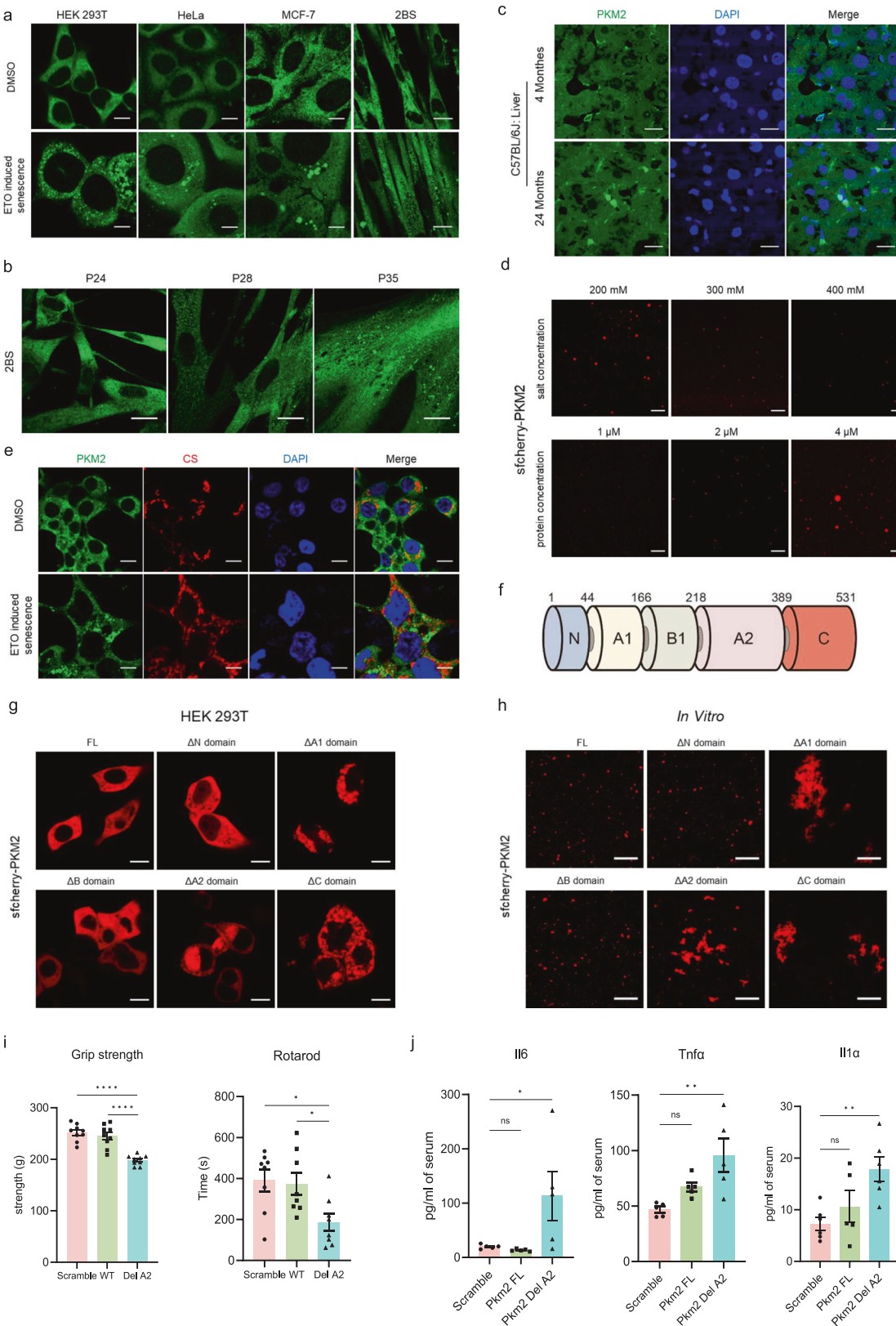

indicated higher degree of aging. The above data demonstrated that PKM2 aggregation is the driving force to inducing cellular senescence or aging in vivo.

## PKM2 aggregates comprise many glycolytic enzymes

PKM2 were capable of aggregating by exogenous expression of sfcherry-PKM2 in HeLa (Fig. 3a) and HEK 293T cells (Supplementary Fig. 4a). To identify the constituent of PKM2 aggregates, sfcherry-PKM2 aggregates were obtained from HeLa cells (Fig. 3a) and sorted using the Fluorescence-Activated Cell Sorting (FACS) technique (Fig. 3b, Supplementary Fig. 4b). The purified PKM2 aggregates were subjected to mass spectrometry (MS) analysis. The high abundance of PKM2 in MS results indicated the reliability of the sorting system. Besides PKM2, PKM2 aggregates were also comprised of many other

**Fig. 2 | PKM2 aggregates accumulate in senescent cells. a** Immunofluorescent imaging of PKM2 in HEK 293T, HeLa, MCF-7, 2BS cells treated with 2 µM etoposide (ETO) for three days. Scale bar, 20 µm for 2BS, 10 µm for the others. **b** Immunofluorescent imaging of PKM2 in fibroblasts 2BS with different passages. Scale bar, 20 µm. **c** Immunofluorescent imaging of PKM2 in frozen section of liver from young and aged mice. Scale bar, 20 µm. **d** Confocal microscopy of purified sfcherry-PKM2 in vitro under different protein concentration and salt concentration. Scale bar, 10 µm. **e** Immunofluorescent imaging of PKM2 and CS (citrate synthase, mitochondria marker) in senescent HEK 293T cells induced by 2 µM etoposide (ETO). Scale bar, 10 µm. **f** Schematic diagram showing the five domains of PKM2. **g** Fluorescent imaging of sfcherry-PKM2 in HEK 293T cells transfected with full length or truncated variants of PKM2. Scale bar, 10 µm. **h** Fluorescent imaging of purified full length or truncated variants of PKM2 in vitro. Scale bar, 10 µm. **i** Tests of grip strength (left, $n = 9$ mice) and rotarod time (right, $n = 8$ mice) were performed in mice overexpressed with wild-type Pkm2 and aggregate-prone Pkm2 mutant (A2 domain deletion) via AAV9 system for 14 weeks. One-way ANOVA was used. $P = 0.0225$ (Scramble vs. Del A2), $P = 0.0379$ (WT vs. Del A2). **j** ELISA measurement of IL6, Tnfα, Il1α concentration in mouse serum of the mice (**i**). $n = 5$ mice, one-way ANOVA was used, $P = 0.9789$ (IL6: Scramble vs. Pkm2 FL), $P = 0.0481$ (IL6: Scramble vs. Pkm2 Del A2); $P = 0.2388$ (Tnfα: Scramble vs. Pkm2 FL), $P = 0.0053$ (Tnfα: Scramble vs. Pkm2 Del A2); $P = 0.5077$ (Il1α: Scramble vs. Pkm2 FL), $P = 0.0080$ (Il1α: Scramble vs. Pkm2 Del A2). NS, not significant, $*P < 0.05$, $**P < 0.01$, $****P < 0.0001$. Error bars represent SEM.

glycolytic enzymes, such as GAPDH, ENO1, PFKP and so on (Fig. 3c). Intriguingly, the reciprocal interaction of these major constituents of PKM2 aggregates had been reported, and PKM2 possesses high propensity for self-assembly according to the algorithm proposed by Chen et al.[27]. This was supplemental evidence to support the existence of PKM2 aggregates (Fig. 3d). The existence of these proteins in PKM2 aggregates were confirmed by Western blot (Fig. 3e) and immunofluorescent imaging (Fig. 3f, g, Supplementary Fig.4c–g). HSPA8 (HSC70) was reported to be the chaperone targeting acetylated PKM2 for chaperone-mediated autophagy (CMA)[28]. Consequently, the partial colocalization of HSPA8 and PKM2 aggregates might count for their accumulation in lysosomes. HK2, which is the first rate-limiting enzyme in the glycolytic pathway, was not identified in the MS results of PKM2 aggregates. Further immunoblotting (Fig. 3e) and immunofluorescent imaging (Supplementary Fig. 4h) analyzes verified the absence of HK2 in PKM2 aggregates. To understand the necessity of PKM2 in the formation of the specific aggregates, HEK 293T cells were transduced with shRNA lentivirus targeting PKM2 followed by monitoring of PKM2-GAPDH aggregates using confocal microscopy. Upon etoposide treatment, PKM2-GAPDH aggregates could be induced in control cells but not in PKM2 knockdown cells (Fig. 3h), which demonstrated that PKM2 was the indispensable constituent of PKM2 aggregates. In conclusion, the components of PKM2 aggregates include not only the indispensable PKM2, but also many other glycolytic enzymes, which indicates the potential role of PKM2 aggregates in glycolysis.

## PKM2 enzymatic activity and glycolytic flux are impaired in senescent cells

We further investigated the direct effects of PKM2 aggregation. The simplest hypothesis is that PKM2 aggregates influence enzymatic activity of PKM2. Etoposide was used to induce senescence in HeLa (Fig. 4a) and HEK 293T cells (Supplementary Fig. 5a) which were then used to measure PKM2 enzymatic activity via PKM2-lactate dehydrogenase (LDH) coupled assay. The slope of NADH fluorescence reflecting PKM2 enzymatic activity significantly decreased upon etoposide-induced senescence (Fig. 4b, Supplementary Fig. 5b). Moreover, as the etoposide treating days extended, PKM2 enzymatic activity gradually weakened in HeLa (Fig. 4c, d) and HEK 293T cells (Supplementary Fig. 5c–f). PKM2 enzymatic activity was also suppressed in senescent HeLa cells induced by Camptothecin (CPT) and 5-Fluorouracil (5-FU) (Supplementary Fig. 5g, h). Besides the PKM2-lactate dehydrogenase (LDH) coupled assay mentioned above, a commercial kit adopting a different principle was also applied to measure PKM2 enzymatic activity and confirmed the same declined activity upon etoposide-induced senescence (Fig. 4e, Supplementary Fig. 5i). Besides DNA damaging-induced senescence, PKM2 enzymatic activity was also impaired in replicative senescent fibroblasts obtained by cell passaging via two independent measuring methods of PKM2 enzymatic activity (Fig. 4f–h). Because the protein amount of PKM2 remained unchanged in these senescent cells (Fig. 4c, g), the decrease of enzymatic activity might be the results of PKM2 aggregation. As PKM2 is one of the three rate-limiting enzymes of glycolysis, defects in

PKM2 enzymatic activity will, more or less, influence glycolytic flux. As expected, production of lactate, which is the terminal product of glycolysis, significantly decreased in etoposide, CPT, or 5-FU-induced senescent cells (Fig. 4i–l, Supplementary Fig. 5j–l) and fibroblasts undergoing replicative senescence (Fig. 4m). These results revealed the decrease of PKM2 enzymatic activity and disturbances of glycolytic flux in senescent cells, accompanying with PKM2 aggregation.

## Dispersing PKM2 aggregates impedes cellular senescence

As PKM2 aggregation impaired PKM2 enzymatic activity and therefore disturbed glycolytic flux and accelerated senescence, we speculated that compounds with the ability to disperse PKM2 aggregation might rescue PKM2 enzymatic activity and alleviate senescent phenotypes. To achieve this goal, we designed a two-step screening program to search for the ideal compounds (Fig. 5a). First, an in-house small molecule library was applied to measure PKM2 enzymatic activity using a PKM2-LDH-coupled assay for PKM2 agonists and antagonists which had higher possibilities to lessen PKM2 aggregation (Supplementary Fig. 6a). These agonists and antagonists could be divided into six categories according to their structures (Supplementary Fig. 6b). Second, the six representative compounds from each category were selected to evaluate their effects on PKM2 aggregation and senescence-associated phenotypes. One of the compounds, K35, was found to be the desired compound in dissipating PKM2 aggregates (Fig. 5b) and alleviating senescence-associated phenotypes (Fig. 5c–f, Supplementary Fig. 6c, d). However, the other five types of PKM2 activators or inhibitors had little effect on lessening PKM2 aggregation or even aggravating it (Supplementary Fig. 6e). For more potent compounds, eight analogs of K35 were evaluated for influence on PKM2 aggregation and senescent phenotypes which led to the identification of K27 as the most efficient compound (Fig. 5b, g, Supplementary Fig. 6f–j). Consequently, we picked K35 and K27 for further investigation. Besides HEK 293T cells, K35 and K27 could also reduce senescent phenotypes induced by etoposide in MCF-7 cells (Supplementary Fig. 6k) and fibroblasts (Fig. 5h, i, Supplementary Fig. 6l). Besides DNA damage-induced senescence, K35 and K27 also ameliorated replicative senescence in fibroblasts (Fig. 5j–l, Supplementary Fig. 7a). The EdU assay indicated that K35 and K27 could rescue repressed cell proliferation by etoposide (Fig. 5m, n, Supplementary Fig. 7b, c). In addition, both K35 and K27 were able to restore the reduced production of lactate due to DNA damage-induced senescence (Fig. 5o, Supplementary Fig. 7d, e) or replicative senescence (Fig. 5p). By measuring PKM2 enzymatic activity, we inferred that the underlying mechanism of these benefits to senescent cells might result from re-activating PKM2 by K35 or K27 (Fig. 5q, r, Supplementary Fig. 7f–h). To clarify the principle of PKM2 activation, we treated HeLa and HEK 293T cells with K35 and K27 but without etoposide, to see if PKM2 activation by these compounds is also seen in non-senescent cells with no PKM2 aggregation. The enzymatic assay indicated that only high concentration of K35 could significantly activate PKM2 in non-senescent cells, while K27 could not (Supplementary Fig. 7i).

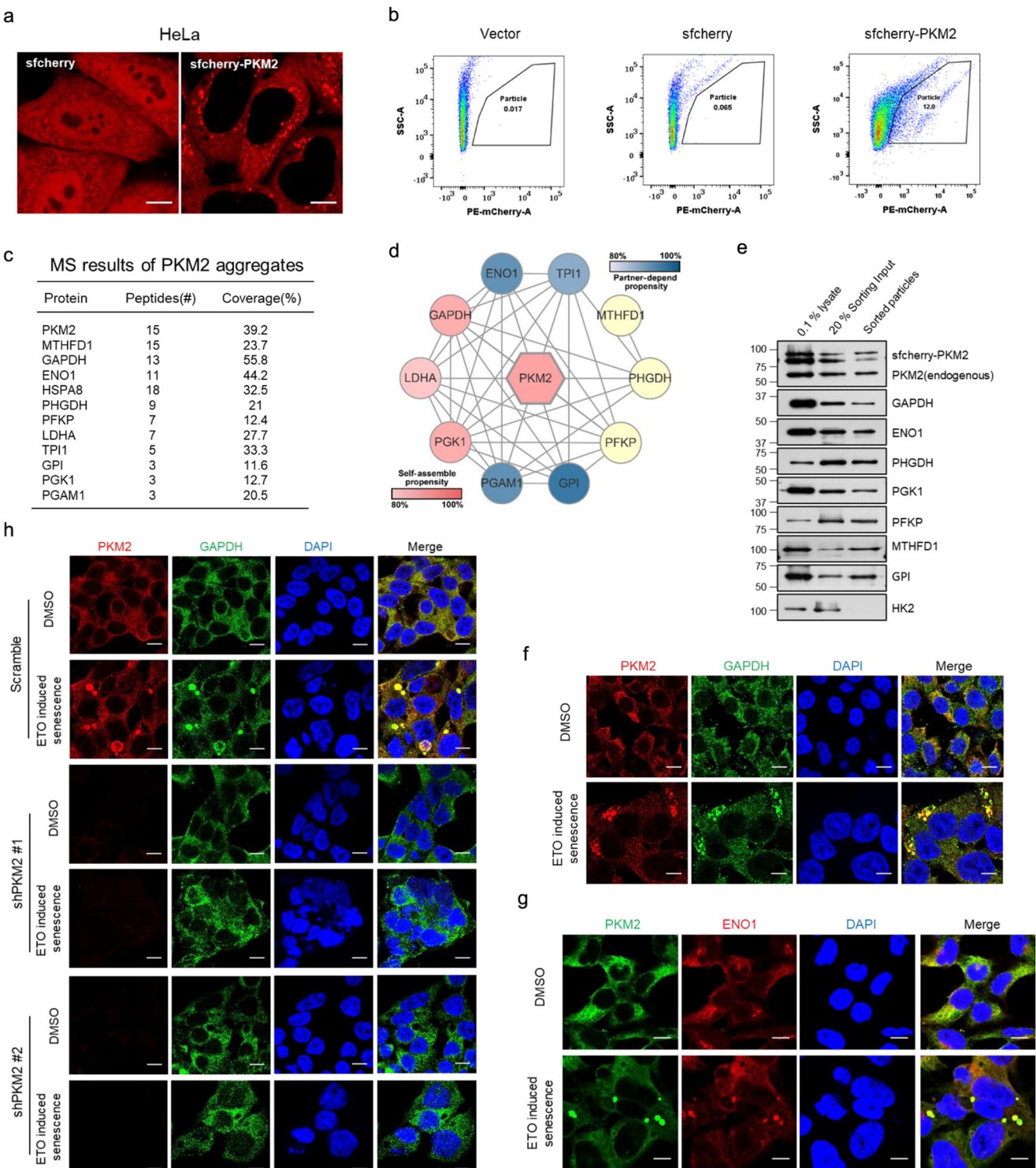

**Fig. 3 | PKM2 aggregates comprise many glycolytic enzymes. a** Confocal microscopy in HeLa cells stably expressed with sfcherry or sfcherry-PKM2. Scale bar, 10 μm. **b** HeLa cells stably expressed with vector, sfcherry, sfcherry-PKM2 were collected and lysed followed by FACS to isolate PKM2 aggregates. **c** The proteins with high abundance in the MS results of PKM2 aggregates isolated by FACS. **d** Schematic view of the PKM2 protein-protein interaction network collected from STRING. Proteins were colored according to the phase separation propensity predicted by PhaSePred. The line between two proteins indicated their interaction gathered from high or low throughput. **e** Immunoblotting of PKM2 and proteins with high abundance in the MS results of PKM2 aggregates. **f, g** Immunofluorescent imaging of PKM2 and GAPDH (**f**) or ENO1 (**g**) in senescent HEK 293T cells induced by 2 μM etoposide (ETO). Scale bar, 10 μm. **h** Immunofluorescent imaging of PKM2 and GAPDH in scramble or PKM2 knockdown HEK 293T cells treated with 2 μM etoposide (ETO). Scale bar, 10 μm. All the above experiments were repeated thrice on separate days with similar results. Source data are provided as a Source Data file.

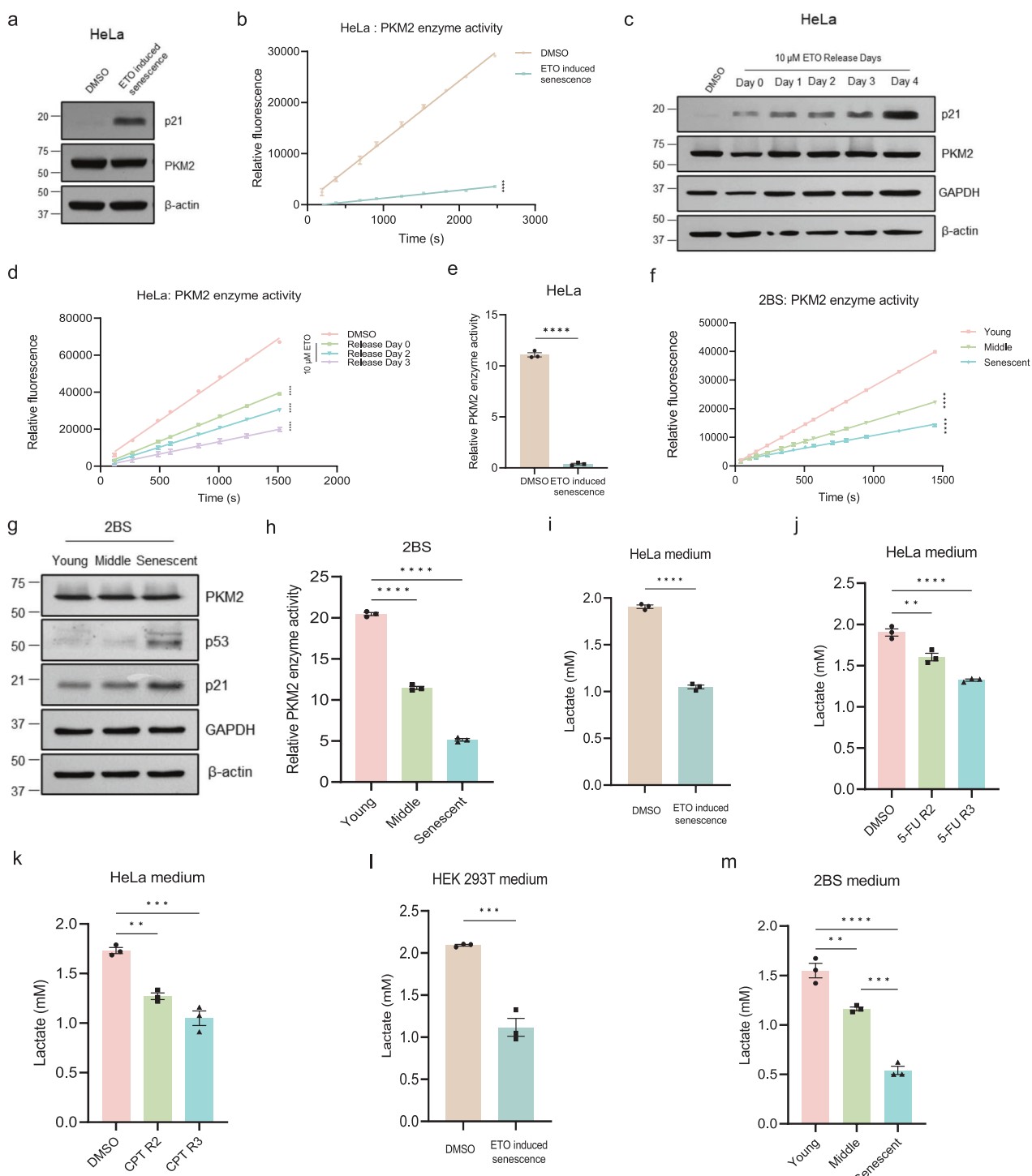

**Fig. 4 | PKM2 enzymatic activity and glycolytic flux are impaired in senescent cells. a**, **b** Immunoblotting of p21, PKM2 (**a**) and measurement of PKM2 enzymatic activity via LDH-coupled kinetic assay (**b**) in HeLa cells treated with 2 μM etoposide (ETO) for three days. $n = 3$ (technical triplicates), Two-tailed unpaired. **c**, **d** Immunoblotting of p21, PKM2 (**c**) and measurement of PKM2 enzymatic activity via LDH-coupled kinetic assay (**d**) in HeLa cells treated with 10 μM ETO for 24 h and release for indicated days. $n = 3$ (technical triplicates), one-way ANOVA. **e** Measurement of PKM2 enzymatic activity with a commercial kit (K709-100) in HeLa cells treated with 2 μM ETO for three days. $n = 3$ (biological replicates), Two-tailed unpaired t-test. **f**–**h** Immunoblotting of p53, p21 and PKM2 (**g**) and measurement of PKM2 enzymatic activity via LDH-coupled kinetic assay (**f**) or using a commercial kit (K709-100) (**h**) in young, middle and senescent fibroblasts 2BS. $n = 3$ (**f**, technical triplicates). $n = 3$ (**h**, biological replicates), one-way ANOVA. **i** Quantification of lactate in the culture medium of HeLa cells treated with 10 μM

ETO and released for three days. $n = 3$ (biological replicates), Two-tailed unpaired t-test. **j** Quantification of lactate in the culture medium of HeLa cells treated with 100 μM 5-Fluorouracil (5-FU) and released for indicated days. n = 3 (biological replicates), one-way ANOVA, $P = 0.0023$ (DMSO vs. 5-FU R2). **k** Quantification of lactate in the culture medium HeLa cells treated with 50 nM Camptothecin (CPT) and released for indicated days. $n = 3$ (biological replicates), one-way ANOVA, $P = 0.0011$ (DMSO vs. CPT R2), $P = 0.0001$ (DMSO vs. CPT R3). **l** Quantification of lactate in the culture medium of HEK 293T cells treated with 10 μM ETO and released for three days. $n = 3$ (biological replicates), Two-tailed unpaired t-test, $P = 0.0008$. **m** Quantification of lactate in culture medium of young, middle and old fibroblasts 2BS. $n = 3$ (biological replicates), one-way ANOVA, $P = 0.0035$ (Young vs. Middle), $P = 0.0003$ (Middle vs. Senescent). **$P < 0.01$, ***$P < 0.001$, ****$P < 0.0001$. Error bars represent SEM. Source data are provided as a Source Data file.

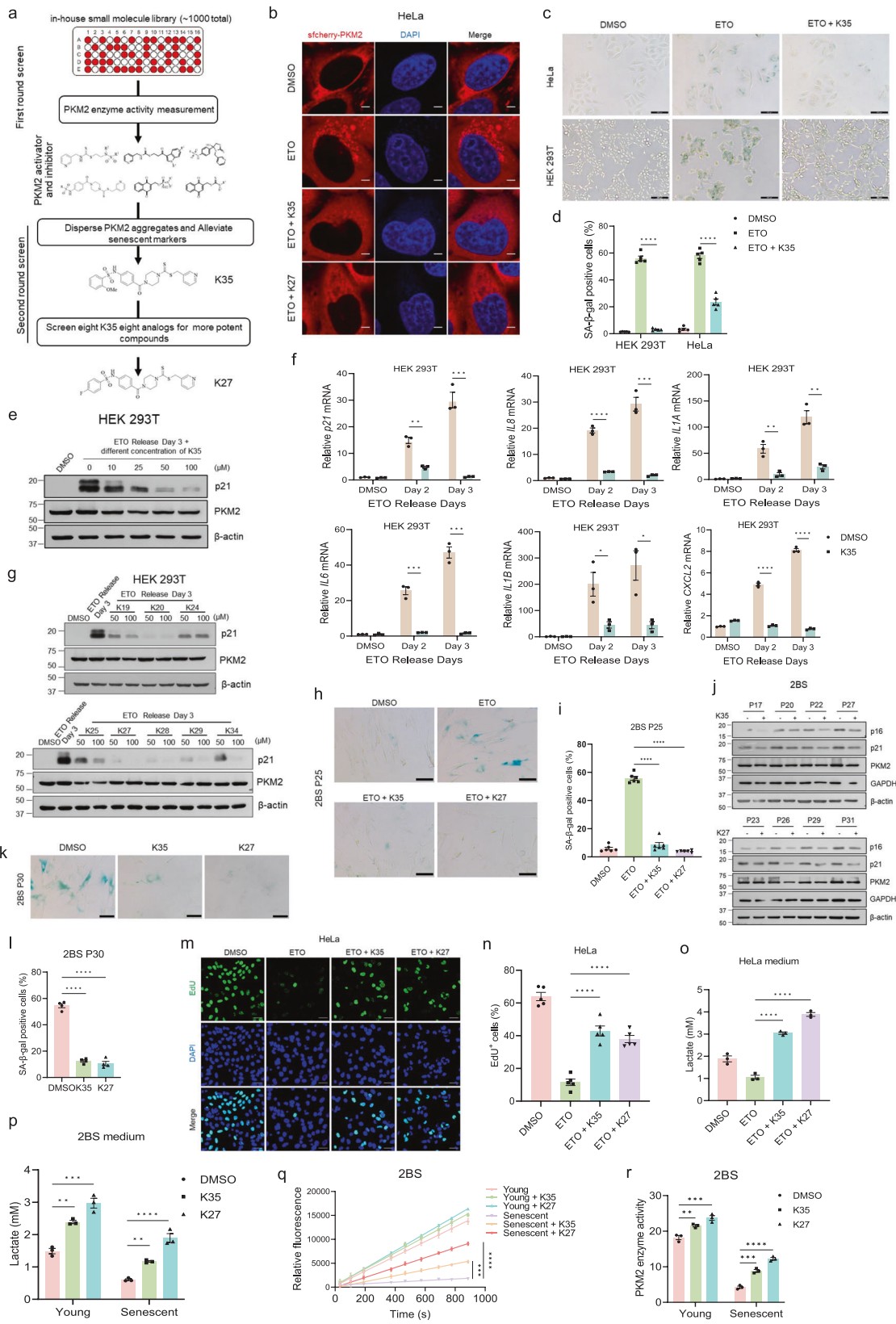

Consequently, K35 might own an aggregate-independent activation of PKM2 activity, while K27 might not. Cell viability assay of young or senescent cells treated with K35 or K27 showed the same trends, which didn't support senolytic effects of K35 or K27 (Supplementary Fig. 7j–l). Molecular docking of PKM2 and K35 or K27 revealed that the two compounds localize at the interface of two PKM2 monomer

(Supplementary Fig. 7m–o). The Microscale Thermophoresis (MST) Analysis of K35/K27 and PKM2 demonstrated that PKM2 could directly bind to K35 or K27 (Supplementary Fig. 7p). Consequently, these results indicated K35 or K27 might disolve PKM2 aggregation by directly binding and remolding the PKM2 structure. Taken together, K35 and its analog K27 were identified to be efficient compounds in

**Fig. 5 | Dispersing PKM2 aggregates impedes cellular senescence. a** Model of screening for compounds dispersing PKM2 aggregates. **b** Immunofluorescent imaging of PKM2 aggregates in HeLa cells stably expressed sfcherry-PKM2 and treated with 2 μM etoposide (ETO) together with DMSO/K35/K27 for two days. Scale bar, 5 μm. **c, d** Representative images (**c**) and quantification (**d**) of SA-β-gal staining in HeLa or HEK 293T cells treated with 10 μM etoposide for 24 h and released for three days together with DMSO or K35 (50 μM). Scale bar, 100 μm. *n* = 5 (randomly captured images), one-way ANOVA. **e** Immunoblotting in HEK 293T cells treated with 2 μM ETO for three days together with concentration gradient of K35. **f** qPCR analysis in HEK 293T cells treated with 10 μM ETO for 24 h and released for indicated days together with DMSO or K35 (50 μM). *n* = 3 (biological replicates), Two-tailed unpaired t-test, *P* = 0.0012 (*p21*: Day 2), *P* = 0.0006 (*p21*: Day 3); *P* = 0.0055 (*IL1A*: Day 2), *P* = 0.0020 (*IL1A*: Day 3); *P* = 0.0004 (*IL6*: Day 2), *P* = 0.0001 (*IL6*: Day 3); *P* = 0.0295 (*IL1B*: Day 2), *P* = 0.0169 (*IL1B*: Day 3). **g** Immunoblotting in HEK 293T cells treated with 10 μM ETO for 24 h and released for three days together with DMSO, K35 or K35 analogs. **h, i** Representative images (**h**) and quantification (**i**) of SA-β-gal staining in fibroblasts 2BS (P25) exposed to 20 μM ETO for 24 h and released for three days together with DMSO, 50 μM K35/K27. Scale bar, 100 μm. *n* = 6 (randomly captured images), one-way ANOVA. **j** Immunoblotting in fibroblasts 2BS constantly treated with DMSO, 25 μM K35/K27. **k, l** Representative images (**k**) and quantification (**l**) of SA-β-gal staining in fibroblasts 2BS cultured with DMSO, 25 μM K35/K27 constantly. Scale bar, 100 μm. *n* = 4 (randomly captured images), one-way ANOVA. **m, n** Representative confocal images (**m**) and quantification (**n**) of EdU staining proliferation assay in HeLa cells treated with 10 μM ETO for 24 h and released for three days together with DMSO, 50 μM K35/K27. Scale bar, 50 μm. *n* = 5 (randomly captured images), one-way ANOVA. **o** Quantification of lactate in the culture medium of HeLa cells exposed to 10 μM ETO for 24 h and released for three days together with DMSO, DMSO, 50 μM K35/K27. *n* = 3 (biological replicates), one-way ANOVA. **p** Quantification of lactate in the culture medium of young or senescent fibroblasts 2BS exposed to DMSO, 25 μM K35/K27. *n* = 3 (biological replicates), one-way ANOVA, *P* = 0.0016 (Young: DMSO vs. K35), *P* = 0.0001 (Young: DMSO vs. K27); *P* = 0.0050 (Senescent: DMSO vs. K35). **q** Measurement of PKM2 enzymatic activity via LDH-coupled kinetic assay in young or senescent fibroblasts 2BS exposed to DMSO, 25 μM K35/K27. *n* = 3 (technical triplicates), one-way ANOVA, *P* = 0.0002 (Senescent vs. Senescent + K35). **r** Measurement of PKM2 enzymatic activity with a commercial kit (K709-100) in young or senescent fibroblasts 2BS exposed to DMSO, 25 μM K35/K27. *n* = 3 (biological replicates), one-way ANOVA, *P* = 0.0090 (Young: DMSO vs. K35), *P* = 0.0007 (Young: DMSO vs. K27); *P* = 0.0002 (Senescent: DMSO vs. K35). *P* < 0.05, **P* < 0.01, ***P* < 0.001, *****P* < 0.0001. Error bars represent SEM. Source data are provided as a Source Data file.

dissolving PKM2 aggregates, restoring PKM2 enzymatic activities and glycolytic flux, and alleviating cellular senescence.

## Compounds K35 and K27 delay aging of naturally and prematurely aged mice

To evaluate the anti-aging effects of K35 or K27 in vivo, a mice model with doxorubicin accelerating aging was established[29] and administered with vehicle, K35, or K27 every two days (Supplementary Fig. 8a). Distribution of the two compounds in blood and brain was measured by quantitative LC-MS (Supplementary Fig. 8b, c). The results indicated that K35 and K27 resisted the digestive tract and were absorbed into the bloodstream. Also, K35 and K27 had the ability to cross the blood-brain and distributed in the brain of mice. To determine the proper dosage, mice were treated with varying dosages of K35 or K27 and monitored key metrics like body weight (Supplementary Fig. 8d) and consumption of food and water (Supplementary Fig. 8e). After intra-gastric administration for ten consecutive days, mice were sacrificed for routine blood tests and serum biochemical examinations. The routine blood tests showed that K35 or K27 had little influence on the number or percentage of various blood cells (Supplementary Fig. 8f), and the content of lactate dehydrogenase (LDH), Alanine aminotransferase (ALT), creatinine (CREA) was not affected by intragastric administration of K35 or K27 (Supplementary Fig. 8g). These results indicate that K35 or K27 had no significant toxicity in C57BL/6 J mice. Intriguingly, K35 or K27 (50 mg/kg) could partially rescue the loss of body weight (Fig. 6a) and significantly prolong the lifespan of the progeroid mice (Fig. 6b). Moreover, the two compounds increased retention time on rotarod and enhanced grip strength of progeroid mice, indicating the recovery of degenerative physical function (Fig. 6c). Biochemical examinations of progeroid mice serum showed that K35 and K27 restored the content of Alanine aminotransferase (ALT) and Aspartate aminotransferase (AST), which are usually used for the evaluation of liver function (Fig. 6d). Besides these global phenotypes, we also evaluated the effects of K35 or K27 on senescent markers. In the mice liver and lung tissue, the upregulated protein levels of p53, p21, and p16 as well as mRNA levels of *p21*, *p16*, and SASP genes due to doxorubicin treatment were reduced by K35 or K27 administration (Fig. 6e–g, Supplementary Fig. 8h, 9a–d). SA-β-gal staining of frozen sections also showed that K35 or K27 could rejuvenate cells from premature aged mice organ triggered by doxorubicin (Fig. 6h, i, Supplementary Fig. 9e–h). In addition, RNA-sequencing analysis showed that K35 and K27 could rescue the expression of senescence or aging-related genes (Supplementary Fig. 9i), which were further validated by Real-time PCR (Supplementary Fig. 9j). After K35

and K27 had been proven efficient in alleviating doxorubicin-induced aging, we further investigated whether the two compounds also function in naturally aged mice. 18-month-old C57BL/6 J mice were divided into three groups and treated with vehicle, K35, or K27 for four months followed by an evaluation of senescent markers, behavior, and lifespan (Supplementary Fig. 10a). Recording mice body weight every week showed nice tolerance of K35 or K27 (Supplementary Fig. 10b). Importantly, K35 and K27 could significantly increase the median lifespan of naturally aged mice (Fig. 6j). After administering for four months, mice treated with K35 or K27 performed significantly better in physical fitness than those treated with vehicle (Fig. 6k). After administering for six months, these mice were sacrificed for lung or liver which was prepared as frozen sections to perform SA-β-gal staining. Intriguingly, K35 and K27 significantly reduced SA-β-gal staining of bronchiole cells and kidney cells of aged mice (Figs. 6l, m, Supplementary Fig. 10c, d). In addition, K35 and K27 both decreased mRNA levels or protein levels of senescent markers in organs of naturally aged mice (Fig. 6n, o). We also measured the concentration of IL6, Tnfα and Il1α in the serum from the naturally aged mice and observed that K35 and K27 both significantly downregulated these SASP factors (Fig. 6p). As expected, K35 and K27 could decrease PKM2 aggregates in liver and brain of aged mice by immunofluorescent imaging of PKM2, indicating that K35 and K27 function well in vivo (Supplementary Fig. 10e–h). In summary, the compounds K35 and K27 mitigated aging phenotypes and extended the lifespan of naturally and prematurely aged mice.

## Discussion

Aging manifests as a progressive decline in normal physiological functions and systemic deterioration across multiple tissues, rendering individuals more susceptible to various diseases, including cancer, neurodegenerative disorders, cardiovascular issues, and metabolic conditions. Central to this process is the intricate balance between energy production and consumption[30]. While the relationship between aging and cellular metabolism has been elucidated to some extent, the precise regulatory mechanisms governing metabolism in senescent cells and aged organisms remain elusive. In this study, we unveiled an original finding: the formation of aberrant multiprotein complexes, termed PKM2 aggregates, comprising PKM2 alongside numerous other glycolytic enzymes, in diverse types of senescent cells and organs from aged mice. Notably, PKM2 aggregates exacerbated senescent phenotypes by, at the very least, impairing PKM2 enzymatic activities and disrupting glycolytic flux. Furthermore, we identified PKM2 aggregate-dissolving compounds, K35 and its analog K27, which

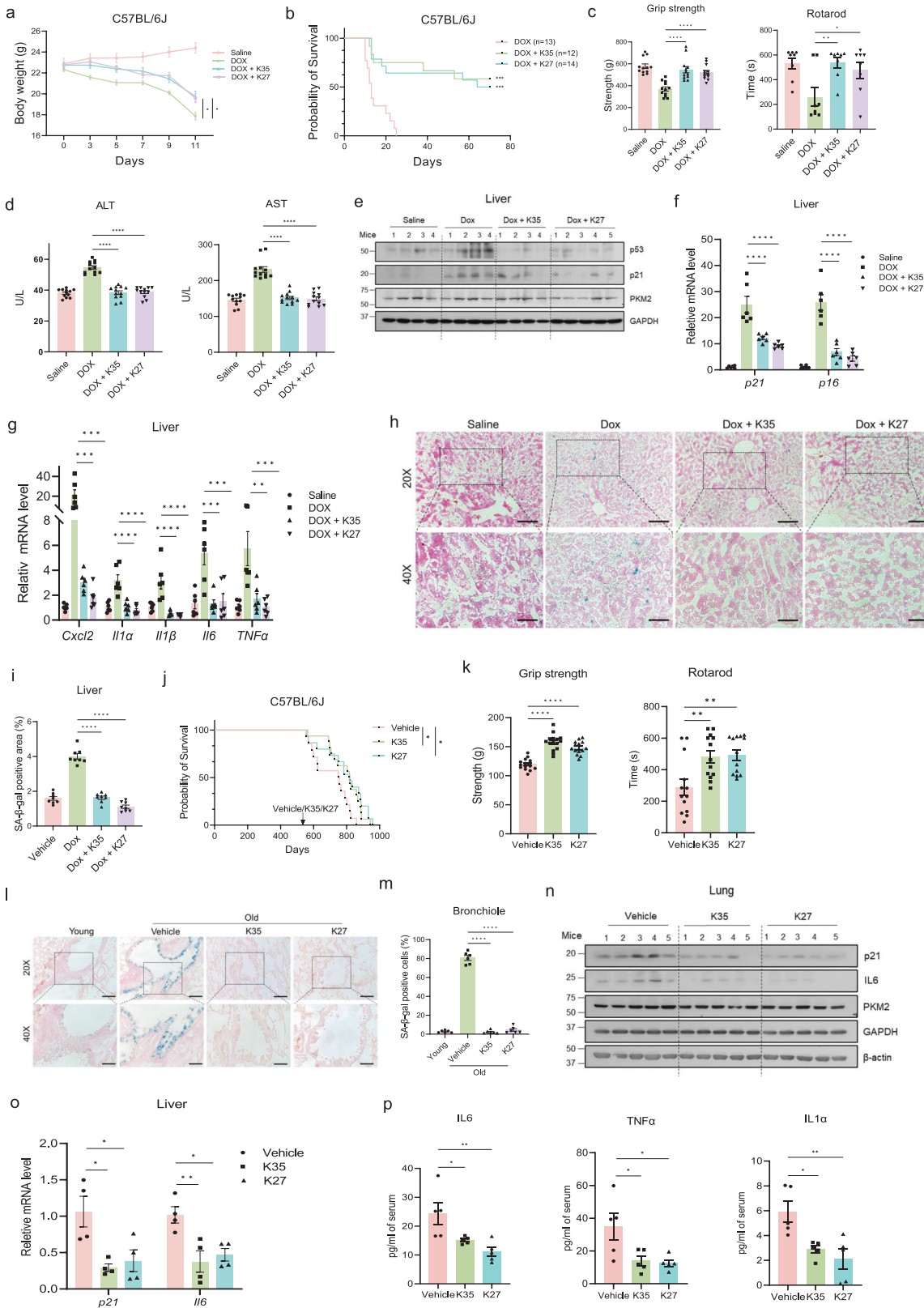

demonstrated the capacity to mitigate cellular senescence and delay aging phenotypes in naturally aged and prematurely aged mice (Supplementary Fig. 10i).

PKM2, a pivotal regulator of cell metabolism, has predominantly garnered attention in cancer research. It is established that PKM2 plays a vital role in cancer development by diverting glycolytic metabolites towards the pentose phosphate pathway (PPP) for nucleotide biosynthesis[31]. Additionally, PKM2 has emerged as a biomarker for various cancers[32-34]. Beyond its well-documented role in cancer, the involvement of PKM2 in the physiological functions of normal tissues, such as cellular senescence, has also been extensively explored. Studies have shown that deletion of PKM2 in colorectal cancer cells

**Fig. 6 | Compounds K35 and K27 delay aging of naturally and prematurely aged mice. a, b** Body weight (**a**) and lifespan (**b**) of C57BL/6 J mice injected intraperitoneally with doxorubicin (10 mg/kg) twice at day 0 and day 7 and intragastrically administered with vehicle or 50 mg/kg K35/K27 every two days from day 2. $n = 7$ mice for K35 groups, $n = 8$ mice for the others, one-way ANOVA (**a**), $P = 0.0122$ (DOX vs. DOX + K35), $P = 0.0179$ (DOX vs. DOX + K27); Log-rank (Mantel-Cox) (**b**), $P = 0.0001$ (DOX vs. DOX + K35), $P = 0.0002$ (DOX vs. DOX + K27). **c** Grip strength (left) and rotarod time (right) of the mice (**a**) were tested at day 20. $n = 12$ mice (left), $n = 8$ mice (right), one-way ANOVA, $P = 0.0064$ (Rotarod: DOX vs. DOX + K35), $P = 0.0361$ (Rotarod: DOX vs. DOX + K27). **d** Examinations of ALT (left) and AST (right) in serum of mice (**a**) at day 20. $n = 12$ mice, one-way ANOVA. **e–g** Immunoblotting (**e**) and qPCR analysis (**f, g**) at day 20. $n = 6$ mice, one-way ANOVA, $P = 0.0008$ (*Cxcl2*: DOX vs. DOX + K35), $P = 0.0004$ (*Cxcl2*: DOX vs. DOX + K27); $P = 0.0004$ (*IL6*: DOX vs. DOX + K35), $P = 0.0007$ (*IL6*: DOX vs. DOX + K27); $P = 0.0026$ (*Tnfα*: DOX vs. DOX + K35), $P = 0.0007$ (*Tnfα*: DOX vs. DOX + K27). **h, i** Representative images (**h**) and quantification(**i**) of SA-β-gal staining in frozen liver tissue sections from mice (**a**) at day 20. Scale bar, 100 μm for 20× magnification images. $n = 8$ mice, one-way ANOVA. **j** Lifespan of 18-month-old C57BL/6 J intragastrically administered with vehicle or 50 mg/kg K35/K27 every five days. $n = 14$ mice per group, log-rank (Mantel-Cox), $P = 0.0153$ (Vehicle vs. K35), $P = 0.0169$ (Vehicle vs. K27). **k** Tests of grip strength (left, $n = 15$ mice) and rotarod time (right, $n = 13$ mice) of mice (**j**) after four months. One-way ANOVA, $P = 0.0035$ (Rotarod: Vehicle vs. K35), $P = 0.0022$ (Rotarod: Vehicle vs. K27). **l, m** Representative images (**l**) and quantification (**m**) of SA-β-gal staining in frozen lung sections from mice (**j**) after six months. Scale bar, 100 μm for 20× magnification images. $n = 6$ mice, one-way ANOVA. **n** Immunoblotting in lung tissue from mice (**j**). **o** qPCR analysis in liver tissue of mice (**j**). $n = 4$ mice, one-way ANOVA, $P = 0.0107$ (*p21*: Vehicle vs. K35), $P = 0.0215$ (*p21*: Vehicle vs. K27); $P = 0.0073$ (*IL6*: Vehicle vs. K35), $P = 0.0172$ (*IL6*: Vehicle vs. K27). **p** ELISA analysis in serum of mice (**j**). $n = 5$ mice, one-way ANOVA, $P = 0.0360$ (IL6: Vehicle vs. K35), $P = 0.0043$ (IL6: Vehicle vs. K27); $P = 0.0257$ (TNFα: Vehicle vs. K35), $P = 0.0185$ (TNFα: Vehicle vs. K27); $P = 0.0176$ (IL1α: Vehicle vs. K35), $P = 0.0042$ (IL1α: Vehicle vs. K27). *$P < 0.05$, **$P < 0.01$, ***$P < 0.001$, ****$P < 0.0001$. Error bars represent SEM. Source data are provided as a Source Data file.

promotes cellular senescence, as evidenced by upregulated p21 protein levels and SA-β-gal activity[26]. Moreover, PKM2 phosphorylates histone H3T11 to attenuate cellular senescence[35]. However, Lunt et al. reported that deletion of PKM2 did not enhance the activity of SA β-gal in young MEFs[36]. On the contrary, PKM2 was identified as a pro-senescent protein whose overexpression promoted expression of p16 and SASP components in a screening of a library containing about 200 kinase[37]. These discrepancies may be reconciled by our theory positing that the active form of PKM2, rather than its abundance, correlates with cellular senescence and aging. Loss of PKM2 impairs glycolytic function, while overexpression of PKM2 augments PKM2 aggregates, both of which promote cellular senescence and aging. Our study underscores the critical importance of the active form of PKM2 in senescence, elucidating the regulatory mechanism of PKM2 function in cellular senescence and aging.

Several studies have highlighted the intimate correlation between glycolysis and cellular senescence. On one hand, glycolysis appears to be more active in senescent cells. Decades ago, studies demonstrated significantly increased glycolytic activity in senescent human diploid fibroblasts, coupled with elevated enzymatic activities of key glycolytic enzymes[38,39]. Conversely, some scientists have observed that higher glycolytic metabolism, characterized by elevated enzymatic activities of glycolytic enzymes, may contribute to the proliferative potential of murine embryonic stem cells compared to senescent MEFs. Additionally, disruption of glycolysis can induce senescence in MEFs, while its overexpression immortalizes them[40]. MEFs at early passages possessed high glycolytic activity, which gradually declined during replicative senescence[41]. Furthermore, glycolytic activity declines in macrophages from aged humans and mice[42]. Moreover, p53, a potent regulator of senescence, is known to suppress glycolysis[43], indicating that glycolysis may be repressed during p53-mediated senescence. Overall, while the relationship between glycolysis and senescence varies depending on cell type, inducer of senescence, and senescence stage, studies consistently suggest that glycolytic flux is disturbed during the onset or progression of senescence, and senescent phenotypes can be mitigated by restoring glycolysis to normal levels. In line with this notion, our results provide a potential mechanism by which senescent cells may downregulate glycolytic flux. Notably, based on this mechanism, we screened drugs capable of restoring glycolysis in senescent cells and aged mice, offering unique insights into metabolic regulation for reprogramming and potential interventions for senescence in cells and aged mice.

Numerous studies have indicated that lysosomes play a crucial role in degrading unfolded proteins, typically through macroautophagy or HSC70-mediated autophagy pathways[44]. Given this, it is reasonable to speculate that lysosomes would target PKM2 aggregates for elimination once they appear. Our results support this

hypothesis; we observed that PKM2 aggregates exhibited increased colocalization with lysosomes in etoposide-induced senescent cells, with the protein levels of PKM2 in lysosomes gradually increasing following etoposide treatment (Fig. 1). Additionally, chaperone HSC70 was detected in PKM2 aggregates through both mass spectrometry analysis of purified PKM2 aggregates (Fig. 3c) and immunofluorescent staining (Supplementary Fig. 4g), suggesting a potential mechanism wherein PKM2 aggregates may enter lysosomes via HSC70-mediated autophagy.

Following a comprehensive screening process, we identified K35 and its analog K27 as compounds capable of dissipating PKM2 aggregates and restoring cellular function. Molecular docking of K35 or K27 with PKM2 revealed the compounds' localization at the interface of two PKM2 monomers (Supplementary Fig. 7m–o), providing insights into their mode of action. However, for a thorough understanding of the mechanisms underlying K35 or K27-mediated dissolution of PKM2 aggregates, urgent resolution of the crystal structure of PKM2 aggregates and their co-crystal is needed. Furthermore, leveraging the co-crystal structure, K35 or K27 could be further optimized to enhance efficacy. Additionally, a high-throughput structure-based virtual screening could be conducted to identify more compounds capable of disrupting PKM2 aggregates and ameliorating cellular senescence. During the screening of targeted compounds to resolve PKM2 aggregates and mitigate cellular senescence, we observed that some PKM2 agonists rescued decreased enzymatic activity in senescent cells but failed to disperse PKM2 aggregates and alleviate senescent phenotypes. This suggests that the adverse effects of PKM2 aggregation extend beyond loss of enzymatic activity; it is likely that the functions of other proteins trapped in PKM2 aggregates are also impaired. Therefore, we posit that K35 and K27, which disperse PKM2 aggregates, not only restore PKM2 enzymatic activity but also preserve the functions of other affected proteins.

In summary, our study elucidates the role of PKM2 aggregates in the aging process. Through the dissolution of PKM2 aggregates, compounds K35 and K27 demonstrated the ability to alleviate aging signatures in both naturally and prematurely aged mice. These findings highlight PKM2 aggregates as a potential target for the development of anti-aging intervention strategies, with K35 and K27 serving as promising lead compounds.

## Methods
### Mice
Animal experiments were proven by the Institutional Animal Care and Use Committee of Peking University Health Science Center. Female BALB/c (8 weeks old) and male C57BL/6 J (8 weeks old) were obtained from the Department of Laboratory Animal Science of Peking University Health Science Center, Beijing. The 18-month-old mice were

purchased from Aniphe Biolab. All these mice were housed according to standards, including constant room temperature and humidity and 12 h: 12 h light cycle. All the mice were free to food (Xietong Pharmaceutical Bio-engineering Co., Ltd, cat#1010002) and drink water. The mice were randomly assigned to each group to minimize the effects of little difference between them.

## Cell lines

HEK 293T, MCF-7 were obtained from American Type Culture Collection (ATCC). HeLa were a kind gift from Qing Chang (Peking university), Zhengfan Jiang (Peking university) respectively. Fibroblasts 2BS were all from Zebin Mao (Peking university). Fibroblasts IMR-90 were all from Tanjun Tong (Peking university). These cells were all cultured in Dulbecco's Modified Eagle's Medium (DMEM) supplemented with 10% Fetal Bovine Serum (FBS), 100 U/mL penicillin and 100 μg/mL streptomycin. All cells were free from mycoplasma contamination.

## In-house compound library

The compound library we selected comes from the research group of Dr. Li Ridong, the co-first author of this article. His research group has been dedicated to the study of PKM2 agonists and PKM inhibitors for many years. Considering that the research content of this article is related to PKM2, the compounds selected in the first round of screening were all PKM2 agonists or PKM inhibitors. The structures of these compounds were verified by nuclear magnetic resonance hydrogen spectroscopy (1H NMR) and nuclear magnetic resonance carbon spectroscopy (13 C NMR), and their purity was greater than 95%.

## Materials and software

qPCR primers are provided in Supplementary Table 1, while other commercial reagents are listed with respective commercial information in Supplementary Table 2.

## Plasmid construction

cDNA of TMEM192-3HA or TMEM192-2Flag was amplified into HBLV vector to generate HBLV-TMEM192-3HA which could stably overexpress in cells. cDNA of sfcherry or sfcherry-PKM2 were amplified by PCR and cloned into HBLV vector to generate HBLV-sfcherry or HBLV-sfcherry-PKM2 for stable overexpress in cells. PKM2-targeted shRNAs were inserted into pLKO.1 for protein knocking down.

## Production of lentiviral particles

HEK 293T cells in 10 cm dishes were transfected with 6 μg pSPAX2, 2 μg pMD2G and 8 μg pLKO.1 plasmid of interested genes using polyethylenimine (PEI). At 48 h posttransfection, the viral supernatant was collected and filtered via the 0.22 μm filter (Millipore). Then the filtered supernatant was concentrated with Lentivirus concentration kit according to manufacturer's instruction and aliquoted into three individual vials. These aliquots were stored in −80 °C freezer until use.

## Lentivirus transduction of cells

Cells were seeded in 6-well plates and infected with lentivirus together with polybrene at 70% confluence. After 48 h, the infected cells were transferred to 6 cm dish and selected with puromycin for one week. These selected cells were collected to test the efficiency of transduction.

## PKM2 aggregating in vitro

sfcherry, sfcherry-PKM2 FL or truncated variants was cloned into pGEX-4T-3, which was expressed in E. coli followed by purification. sfcherry and sfcherry-PKM2 were eluted in ultrapure water containing reduced glutathione (50 mM). To determine the influence of salt concentration on PKM2 aggregating, the same amount of sfcherry or sfcherry-PKM2 was mixed with salt solution whose salt concentration is known to reach the indicated salt concentration. Then, the mixture was added onto slides and covered by cover glasses followed by confocal imaging.

## Measurement of PKM2 enzyme activity

In this manuscript, we used two methods to measure PKM2 enzyme activity. One of them, called LDH-coupled kinetic assay, reflects PKM2 enzyme activity by determining the slopes of NADH fluorescence decrease catalyzed by LDH. After treatment, cells were collected and lysed with lysis buffer for 40 min followed by ultrasonication (30% power; 1 s on, 3 s off) for 30 s. Protein concentration was determined with Bradford (Bio-Rad, #500 0205). According to the protein concentration, the protein solution was normalized to 1 μg/μL. Then, 2 μL protein solution was mixed with 50 μL stock 1 solution (45 μL buffer, 5 μL 10 mM PEP) and incubated in room temperature for 5 min. Stock 2 solution (34 μL buffer, 5 μL 80 mM ADP, 5 μL 4 mM NADH, 1 μL LDH enzyme) was added to stock 1 solution immediately and measured NADH fluorescence (excitation wavelength: 340 nm, emission wavelength: 460 nm) in Corning® 96-well Flat Clear Bottom Black Polystyrene TC-treated Microplates (Corning, #3904) with micro-plate reader (POLARstar Omega). The decreasing slope of NADH fluorescence reflects the enzyme activity of PKM2. Another method measuring PKM2 enzyme activity was performed with a commercial kit (Biovision, #709-100). In the assay, PEP and ADP were catalyzed by PKM2 to generate pyruvate which is oxidized by pyruvate oxidase to produce color ($\lambda = 570$ nm). By measuring changes of absorbance at 570 nm, PKM2 enzyme activity could be determined.

## Measurement of lactate in medium

1 mL cell culture medium was collected from dishes and then centrifuged at 4 °C for 3 min at $1000 \times g$. 500 μL supernatant was ultrafiltrated using 10 kDa Centrifugal Filter (UFC5010). The corresponding cells were also collected and lysed for determination of protein concentration. Then, the medium in the collection tube was adjusted according to the protein concentration. Finally, the adjusted medium was used for determination of lactate using CheKine™ Micro Lactate Assay Kit (KTB1100) according to the manufacturers' instructions.

## Silver staining

The SDS-PAGE gel was fixed in 50 ml 50% ethanol + 10% acetic acid at 4 °C overnight. Then the gel was washed for 20 min (three times) with 30% ethanol followed by incubation with 50 mL sensitizing solution (500 μL 0.2% $Na_2S_2O_3$, 15 mL absolute ethanol, 3.4 g Sodium acetate) for 40 min at room temperature. Later, the gel was rinsed with water for 10 min (two times) and incubated with 100 mL silver staining buffer (0.3 g AgNO3, 100 μL 37% formaldehyde) followed by washing with water for 10 s (three times). Finally, the gel was incubated with 100 mL developing buffer (3 g $Na_2CO_3$, 1 mL 0.2% $Na_2S_2O_3$, 50 μL 37% formaldehyde) before protein bands was clear.

## RNA isolation and Real-time quantitative PCR

TRI Reagent (Sigma-Aldrich) was used to extract total RNA from cells or tumors. Then, 2 μg total RNA was reversely transcribed into cDNA using the cDNA Synthesis Kit (Yeasen, 11123ES10). The cDNA was diluted 3-10-fold and mixed with qPCR SYBR Green Master Mix (Yeasen, 11202ES03) and coupled primer pairs. The Real-time quantitative PCR was performed on 7500 Real-time PCR System (Applied Biosystems).

## Lysosome immunoprecipitation

The lysosome localized protein TMEM192-3HA was stably overexpressed in HEK 293T cells. These cells were treated cells with 5-FU for different days and the cells were harvested and homogenized to collect the mixture containing intact lysosomes by centrifuging at

$1000 \times g$. Anti-HA Magnetic Beads (Pierce, #88836) were incubated with the mixture and rotate for 50 min at 4 °C. The beads were washed with KPBS (136 mM KCl and 10 mM KH2PO4 pH 7.25) using DynaMag™ (Invitrogen, #12321D) and eluted with RIPA.

## PKM2 aggregates isolation by Flow cytometry

HeLa cells with stable expression of vector, sfcherry or sfcherry-PKM2 were collected and lysed with lysis buffer (50 mM Tris-HCl pH 7.9, 137 mM NaCl, 1% Triton X-100, 0.2% Sarkosyl, 1 mM Na3VO4, and 10% glycerol) for 40 min followed by ultrasonication (30% power; 1 s on, 3 s off) for 20 s. Then, the lysate was centrifuged at $1000 \times g$, 4 °C for 3 min. The supernatant was transferred to a new tube followed by centrifuging at $5000 \times g$, 4 °C for 10 min. Next, discarding the supernatant and washing the pellets with lysis buffer by gently pipetting followed by centrifuging at $5000 \times g$, 4 °C for 10 min followed by washing the pellets and centrifuging again. Finally, resuspending the pellets with lysis buffer and sorting with flow cytometry (Symphony S6).

## Immunofluorescent staining

For imaging of fixed cells, cells were seeded in 6-well plates in which cover glasses were put inside in advance. After treatment, the cells on the cover glasses were fixed in 4% Paraformaldehyde (PFA) and permeabilized by 0.3% Triton X-100 in PBS. The cells were blocked with 3% goat serum and 5% BSA in PBS followed by primary antibodies incubation overnight at 4 °C. After washing with PBS, cells were incubated with secondary antibody including Alexa Fluor™ 488 (Invitrogen, A-21206) and Alexa Fluor™ 594 (Invitrogen, A-21203) for 1 h at 37 °C. Finally, cells were incubated with DAPI and mounted to slides.

For imaging of living cells, cells were seeded in glass bottom cell culture dishes. Before imaging, cells were incubated with Hoechst 33342 (Solarbio, C0031) for 30 min in CO2 incubator.

For immunofluorescent staining in organ sections, tissues were fixed in fixed in 4% Paraformaldehyde (PFA), dehydrated in 30% sucrose and then embedded in optimal cutting temperature compound (OCT, #4583) which were immediately frozen at −80 °C freezer until use. The tumor blocks were cryo-sectioned to 10 μm sections using Leica Cryostat (Leica CM1950). Tumor sections were washed thrice with TBS to wash residual OCT compound followed by antigen retrieval solution treatment (Solarbio, C1035). Then the tissue sections were blocked in 1% BSA, 10% goat serum in TBS for 1 h at 37 °C and incubated with primary antibodies against: anti-p16 (sc-1661) or anti-p21 (sc-6246) overnight at 4 °C followed by secondary antibodies incubation with Alexa Fluor™ 594 (Invitrogen, A-21203) for 1 h at 37 °C. Finally, tissue sections were incubated with DAPI for 15 min and mounted to slides. All immunofluorescent imaging was performed on ZEISS LSM 880.

## DNA damage-induced senescence

HEK 293T, HeLa, MCF-7 cells were treated with 2 μM etoposide for 1–3 days three days to induce senescence with different degree. HEK 293T, HeLa cells were treated with 10 μM etoposide and released for 0–4 days to induce senescence at different level. MCF-7, 2BS cells were treated with 20 μM etoposide and released for 0–4 days to induce senescence at different level. HeLa, HEK 293T cells were treated with 100 μM 5-FU and released for 2–3 days to induce senescence at different level. HeLa, HEK 293T cells were treated with 50 nM or 25 nM CPT respectively and released for 2–3 days to induce senescence at different level.

## Cytotoxicity assays with CCK-8

Young or senescent fibroblasts 2BS were seeded in 96-well dishes. When the confluence reached 70%, the 2BS cells were treated with different concentration of K35 or K27 for three days. Then, cell viability was determined by CCK-8 kit (CK04) according to the instruction of the manufacturer.

## EdU cell proliferation assy

HeLa or HEK 293T cells were seeded in six-well dishes in which cover glasses were put inside in advance. After the cells were adherent to the cove glasses, cells were treated with 10 μM etoposide for 24 h and released for three days to induce cell cycle arrest in combination with DMSO, 50 μM K35, K27 followed by EdU staining proliferation assay (BeyoClick™ EdU-488) according to the instruction of the manufacture.

## SA-β-gal staining

For SA-β-gal of cultured cells in dishes, cells were seeded in six-well dishes and cultured with exposure to indicated compounds. After the treatment, cells were fixed and stained using SA-β-gal staining kit (Cell Signaling, #9860).

For SA-β-gal staining in the frozen sections, sections were dried at room temperature followed by fixing and staining with SA-β-gal staining kit (Cell Signaling, #9860). After staining for about 12 h, the sections were stained with eosin (Servicebio, G1002) for 60 s and rinsed under running water for 1 min. Then the sections were dehydrated successively in 80%, 95%, 100% ethanol for 10 s and cleared in xylene for two times (5 min per time). After the xylene on the slides volatilize, a cover glass was mounted to the slides.

The SA-β-gal staining cells were observed under a microscope. Images of cultured cells were taken under 20× objective lens. Images of frozen sections were taken under 20X or 40X objective lens.

## AAV transduction in mice

The plasmids of scramble, Pkm2 (mouse) and Pkm2 (mouse) with A2 domain deletion were constructed followed by expression validation by Real-time PCR. Then these plasmids were packaged using AAV9 virus particles. Thirty 13-month-old mice were randomly and evenly separated into three groups. Then about $1 \times 10^{12}$ vg virus were injected in each mouse through tail vein followed by subsequent evaluation.

## Accelerating aging mouse model

Male C57BL/6 J (8 weeks) were intraperitoneally injected with 10 mg/kg doxorubicin (MCE, HY-15142) at day 0 and day 7. From day 2, these premature mice were intragastrically administrated with vehicle, K35 (50 mg/kg) or K27 (50 mg/kg) every two days.

## Toxicity testing in vivo

Male and female C57BL/6 J (8 weeks) were intragastrically administrated with vehicle or different dosage of K35 for ten consecutive days. During the treatment, body weight of mice as well as the whole consumption of food and water were recorded. Because the mice from the same group were housed in one cage, the food or water consumption results had no error bar. At the last day, the mice were sacrificed for routine blood tests and serum biochemical examinations. For routine blood tests, 50–100 μL fresh blood were collected in tubes with EDTA pre-treating. The blood samples were analyzed by hematology analyzer (Hemavet 950FS). For serum biochemical examinations, about 500 μL blood were collected in 1.5 mL tubes and put aside at 37°C for 2 h followed by centrifuging at $1000 \times g$ for 10 min to obtain serum. About 200 μL serum were divided into several aliquots for the analysis of lactate dehydrogenase (LDH), Alanine aminotransferase (ALT), creatinine (CREA) by chemistry analyzer (Mindray BS-180).

## Quantification of K35 or K27 in vivo

Mice were administered intragastrically with K35 (50 mg/kg) or K27 (50 mg/kg). After 0,1,3,6,12,24 h, mice were sacrificed to obtain blood and brain. The blood was put aside at 37°C for 2 h followed by centrifuging at $1000 \times g$ for 10 min to obtain serum. The brain was homogenized and sonicated in 500 μL saline followed by centrifuging

at 15,000 rpm for 20 min to obtain supernatant. 45 μL serum or brain lysate was subjected to quantification of K35 or K27 by LC-MS assay. The concentration of K35 or K27 was calculated according to the calibration curve. A UPLC BEH C18 column (50 × 2.1 mm, 1.7 μm, Waters, Milford, MA, USA) was employed for the chromatographic separation of K35, K27 and internal standard (IS) in a Dionex UltiMate 3000 Ultra–HPLC system (Thermo, San Jose, CA, USA). An API 4000 QTRAP mass spectrometer (AB SCIEX, Foster City, CA, USA) was employed in the mass spectrometric analysis.

### Measurement of food or water intake of mice
Each mouse was raised in a separate cage. For each mouse, the supply and remains of food or water were weighed during the experimental period. The food or water intake of each mouse was calculated by subtracting the remains from the supply.

### Rotarod test
Mice were trained to perform rotarod test for three consecutive days at 20 r/min. Mice were placed on the rod of rotarod machine (JINA-NYIYAN, YLS-4C). Then the rod gradually accelerated to 20 r/min. The falling time of each mouse was recorded. The tests were repeated on next two days. The final results were the average of the three trails. Time below 60 s were discarded. When the time reaches to 600 s, the trail was stopped by the researcher.

### Grip strength test
Mice were put on the grid of grip strength meter (Sansbio, SA417). It must be assured that mice grip the grid with all their front paws. The maximum force (g) was recorded when tail of mouse was grasped and pulled back. This trail was repeated fifteen times consecutively. The final results were the average of fifteen trails.

### ELISA analysis
About 600 μL blood of mice were collected in 1.5 ml tubes followed by clotting at room temperature for 2 h, and then centrifuged at 4 °C for 15 min at 1000 × g to obtain serum. Concentration of mouse IL1α, TNFα and IL6 were measured using Mouse Interleukin 1α (IL1α) ELISA Kit (CUSABIO, CSB-E04621m), Mouse Tumor Necrosis Factor α (TNFα) ELISA Kit (CUSABIO, CSB-E04741m) kit, and Mouse IL6 (Interleukin-6) ELISA Kit (Invitrogen, KMC0061) according to the manufacturers' instructions.

### Molecular docking
The molecular docking of PKM2 with K35 or K27 was processed using the Open-Source Autodock vina from Scripps Research Institute. At first, it was pre-processed using Schrodinger Maestro from Peking University and AutoDock Tools 1.5.6 rc3. Then, it was assessed using Schrodinger Maestro, and pictured using PyMOL 1.3.× afterwards. The X-ray crystal structure of PKM2 (PDB CODE:4G1N) was obtained from the PDB data bank. Receptor preparation: All solvent molecules or any other hetero molecules were removed. All HETATM and hydrogen atoms were deleted, and the crystal cell was removed. Alternative conformers of residues were also deleted to retain one set. The receptor was further processed via AutoDock Tools 1.5.6 rc3. Polar hydrogen and Kollman charge were added to give the prepared protein at last. Ligand preparation: Polar hydrogen and Gasteiger charge was given, and there is no further modification on the ligand torsion tree. Molecular docking: Parameter selection was made via AutoDock Tools 1.5.6 rc3. All remaining parameters were kept as default.

### Microscale thermophoresis analysis
The binding affinity between K27 or K35 and PKM2 was measured by NanoTemper Monolith NT.115 instrument (NanoTemper Technologies, Germany). PKM2 was firstly adjusted to a concentration of 20 μM and then labeled with Monolith NT.115 Protein Labeling Kit RED-NHS

(NanoTemper Technologies, Germany) according to the protocol of the manufacture. After the protein labeling experiment was completed, PKM2 was diluted with binding buffer (50 mM tri-HCl, pH 7.5, 10 mM KCl, 5 mM MgCl2 and 0.005% Tween 20) to ensure that the fluorescence intensity of PKM2 during the MST assay was about 500 RU. In this assay, the final concentration of PKM2 after mixing with the compound was 100 nM. In the following, the compounds were serially diluted in binding buffer (16 points, 1:2 dilutions, from 0.2 mM for K27 or from 0.5 mM for K35), then mixed and incubated with an equal volume of the diluted PKM2 at room temperature for 5 min. After incubation, the samples were loaded into the premium treated capillaries and measured in NanoTemper Monolith NT.115 instrument (NanoTemper Technologies, Germany). The KD values were fitted by NanoTemper Monolith affinity software (NanoTemper Technologies, Germany) using 1:1 binding mode.

### GSEA analysis
We performed the single-sample gene set enrichment analysis (ssGSEA) by matching core components in the glycolysis pathway against proteins collected from the lysosome mass spectrometry (MS) experiment. Proteins were ranked by fold change values derived from Mascot, and the P-value was calculated with a 1000-round permutation.

### Quantification of immunofluorescent images
Several immunofluorescent images were taken randomly. The number of total cells were quantified by counting the number of cell nucleus using Image J. The cells with positive signals were counted by an independent researcher. Then the percentage of cells with positive signals were calculated.

### Statistics and Reproducibility
Quantitative data are presented as the means ± SEM. Prism 9 (GraphPad) was used for the statistical analysis. $P < 0.05$ was considered statistically significant. Information on the statistical tests used, precise P values, sample sizes, and n values are provided in each figure legends. All experiments were repeated at least three times with similar results.

### Reporting summary
Further information on research design is available in the Nature Portfolio Reporting Summary linked to this article.

## Data availability
All related raw sequencing and processed data have been deposited at the Gene Expression Omnibus under accession number GSE268773. All other data are available in the main text or in the supplemental information. The uncropped and unprocessed scans of all the blots are supplied in the Source Data file. Source data are provided with this paper.

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

## Acknowledgements

We thank Zebin Mao (Peking university) for providing us cell lines; Yongpan An (Peking University) for the assistance of rotarod tests. John Luo for manuscript editing assistance. This work was supported by grants from National Natural Science Foundation of China (82172959, 81874147, 81671389 to J.L. and T2325003 to T.L.).

## Author contributions

J.B., T.L., X.L. and J.L. designed the study. J.C aided in animal experiments. R.L. was responsible for synthesis of compounds. J.B. performed most of the experiments with the help from Y.L. and C.S. Z.C drew the model graphs. T.Z. aided in ELISA tests. Z.T. performed the MS analysis of purified PKM2 aggregates. L.Z. and L.S. gave assistance to J.B. on flow cytometry assay. J.W and Y.W help on behavioral experiments.

## Competing interests

The authors declare no competing interests.
