## [Peer Review File · Nature Communications]

PKM2 aggregation drives metabolism reprogramming during aging processREVIEWERS' COMMENTS:

Reviewer #1 (Remarks to the Author):

The authors reported that PKM2 aggregation drives cell senescence and mice aging. Meanwhile the authors identified two activators which can alleviate PKM2 aggregation and delay aging-related phenotype. The effect of two activators on delaying the aging phenotype is quite significant and can be used as potential anti-aging drugs. However, it lacks sufficient direct evidence to prove the effect of PKM2 aggregation on cell senescence. It is confusing whether two activators influence the aggregation or enzyme activity directly. And it lacks connection between mechanism and phenotype. Overall, the materials and techniques here compromise the novelty. And there are some confusing points need to be explained by the authors.

1. Why does the author use cervical cancer cells HeLa and HEK293T for mechanism research at the cellular level, while focusing on the liver, lungs, and muscles at the individual level? What is the connection between these indicators?

2. Whether PKM2 aggregation only occur during DNA damage-induced senescence? Would PKM2 aggregation occur during ROS-induced or oncogene-induced senescence?

3. At the animal level, why does the author not use ETO to induce aging as same as at the cellular level? Otherwise, use both ETO and doxorubicin at the cellular and animal level? The induction should be the same.

4. Through lysosome-MS, the authors found PKM2 and GPI were enriched in lysosomes during the etoposide-induced senescent HEK293T. But why the authors aimed at PKM2 rather than GPI? How to distinguish between PKM2 and its isoenzyme PKM1 through MS? What role does PKM1 play in etoposide-induced senescence?

5. It has reported that some proteins, eg. SIRT1(Xu et al., 2020) and PKM2(Wu et al., 2023) undergo degradation during cell senescence, so why does PKM2 aggregate in lysosomes and not be degraded? We have reported that the expression of PKM2 in etoposide-induced

senescent cells was down-regulated (Wu et al., 2023), here Fig1d showed no change, whether it was due to differences in cell type or something else?

6. In fig1e, the co-location between PKM2 and lysosome is not very well, so Why? It seems that the level of PKM2 in cytoplasm is increased during ETO-induced HEK-293T, nuclear cytoplasmic separation assays should be applied. I doubt that the quantitative method shown by Fig1e-f can be replaced with the area where PKM2 overlaps lysosome vs. the total area occupied by PKM2.

7. Fig2i-j images are not clear, please provide high-quality images, and the images are inconsistent with quantitative data.

8. Fig3h showed that GAPDH showed aggregation in etoposide-induced senescent cells, and studies showed that the loss of PKM2 would induce cellular senescence, so why GAPDH in PKM2-knockdown cells showed the same distribution state as the control group without etoposide, please explain. In Fig3h it seems that the aggregation of GAPDH depends on PKM2, so does the aggregation of other glycolytic metabolic enzymes (ENO1, GPI, LDH or PFKFB3) also depend on PKM2? Why can PKM2 drive the aggregation of other metabolic enzymes?

9. The expression of PKM2 shown by Fig3f is down regulated in etoposide-induced senescent cells, which is clearly different from the results in Fig1d and Fig3h, and the authors should give an explain. Why is the PKM2 aggregation state shown by Fig2e and Fig3h so different?

10. In Fig5g, why the p21 is higher in ETO Release Day3+10mM K24 than in ETO Release Day3?

11. The growth advantages of HeLa cells showed by the treatment of K35 and K27 in Fig5n-o are inconsistent with the lactate content in the medium.

12. Supplementary Fig.S1h, The knockdown efficiency of PKM2 is inconsistent with the previous immunofluorescence, and the knockdown efficiency of #2 is not good, resulting in the aging phenotype is more pronounced than #1 in Supplementary Fig.S4l, why?

13. Supplementary Fig. S4l showed CPT-induced aging with a decrease in lactate levels in HEK 293T medium is not significantly compared to etoposide.

14. In line 871, "ETDA" should be corrected to "EDTA".

Reviewer #2 (Remarks to the Author):

In this work, Bie and colleagues perform lysosome purification followed by proteomics analysis in senescent cells, and identified many glycolytic proteins in senescent cells lysosomes. In particular, they focus on the formation of aggregates of pyruvate kinase M2 (PKM2) that also carry other glycolytic proteins, and are associated to decreased PKM2 activity and glycolytic flux. Authors then perform a drug screening identifying two related compounds (K35 and K27) that disperse PKM2 aggregates and decrease senescent markers in vitro. They then test these compounds in a model of chemotherapy stress and in naturally aged mice, and again show decreased senescence markers and improved survival and neuromuscular performance in K35- and K27-treated mice.

In general, the work is interesting and novel, showing a possible new mechanism of metabolic evolution of cells as they age/senesce that can help explaining the metabolic shift in this physiological process. There are however several points unclear in the description of the results and their interpretation, specially in the last part of the manuscript, that must be clarified before publication.

General remarks:

- Letter size in many figure panels is too small and hard to read.
- I cannot see a proper Statistical analysis report in the Methods section. In particular, I can see in the Figure legends that two-tailed Student t-test was performed in several figures where several groups are compared (Figure 1f, 2j, 4h-m, 5d-r, 5c-p), and a one-way ANOVA should have been used; and no statistical analysis method is indicated for Figures 6a (two-way ANOVA would fit) and 6b (logrank test). Please re-do these analyses and use an appropriate test for each panel, indicating it in the Figure legends and in a complete Statistical analysis sub-section in the Methods section.
- Authors claim that compounds K35 and K27 reduce senescence markers in many cells, including p21 and/or p16 protein, SASP-associated cytokine transcription and SA-b-Gal. Is

this decrease due to senescence reversal, or due to senescence cell death (senolytic effects)? In principle, senescence is considered a very stable cellular status (irreversible for many), and this point is rather controversial. Comparison of cell viability between proliferating and senescent cells treated with vehicle or K35/K27 but without chemotherapy administration would clarify this point.

- The Methods section is confusing, alternating in vitro, in vivo and in silico data. Please, group the methods to facilitate reading.

- No effort is made to evaluate pharmacological parameters of K35 and K27. Bioavailability and tissue distribution of these compounds should be performed, including their ability to cross the blood-brain barrier. This is specially important since these compounds are administered intragastrically, and it is not certain if they resist the digestive tract or are absorbed into the bloodstream.

Major Comments:

- Figure S1: there is HA-tagged TMEM192 in the ER compartment, but no mention is done in the text or the figure legends of this apparent impurity of the IP. Please comment.

- In the Methods section for lysosome IP, authors indicate that HeLa cells stably expressed TMEM192-HA; but in the figures, authors indicated 293T cells. Please verify.

- Figure 1d: it seems that the b-actin loading control of the input blots to the right increased with time of treatment. Please quantify the increase in the HA-IP proteins relative to b-actin loading control, using replicates for each time point.

- Figure 1e, f and S1f: please indicate what dye/marker was used for the identification of lysosomes, and how was the quantification of the merge between PKM2/GPI and lysosomes performed. I cannot see this information in the Methods section. Also, quantify the colocalized GPI-lysosomes for Figure S1f, as shown for PKM2 in Figure 1f.

- Figure S1g-h: please, indicate the exact sequences of the shPKM2 used in these figures.

Also, please quantify, with an appropriate number of replicates (≥ 3), the alleged increase in SA-bGal (Figure S1g) and of p21 (figure S1h) in the shPKM2-infected cells. In particular,

shown images of the SA-bGal staining are not very clear: very few cells are shown for the DMSO baseline controls for the shPKM2-infected cells, showing already high SA-bGal staining. Quantification of large number of cells, relative to SA-bGal-positive cells, should be provided to prove an increase in senescence. Also, since GAPDH is altered in senescence, I would recommend using a different loading control for WB in Figure S1h, such as b-actin, as in Figure 1d.

- Figure S2f: aggregation of sfcherry-PKM2 is not very clear in these images, with some aggregation (stronger staining points) also visible at T0 in sfcherry-PKM2. Please provide clearer images, or quantify the aggregation.

- Figure 2d and S2g: please clarify which cells were used to overexpress sfcherry-PKM2, and how was the purification performed. In the Methods section, only flow cytometry analysis of sfcherry-PKM2 is indicated. Also, indicate how different salt concentrations were reached and monitored.

- Figure 2e: authors claim that “the confocal images (Fig. 1e) showed that not all PKM2 aggregates merged with lysosomes”. However, no lysosome marker is used in these IF images, so this claim is not appropriate. Also, please quantify PKM2 and lysosome or mitochondria co-localization, as done before with Figure 1f, to generate a robust piece of data. Finally, please indicate in the figure legend what does the CS (citrate synthase) marker for mitochondria identification stand for.

- Figure S2h: please quantify PKM2 and stress granule co-localization, as done before with Figure 1f, to generate a robust piece of data.

- Figure S2j: please clarify why the different patterns of Coomassie staining of the truncate mutants of PKM2 illustrate differential aggregate formation in vitro. Also, as requested for Figure 2d and S2g, clarify the protocol followed to obtain these cherry-PKM2 aggregates in vitro.

- In Figure 3h, no PKM2 staining is observed in the IF of shPKM2 #1 and #2. However, in Figure S1h, a significant amount of PKM2 was observed by WB, specially in cells transduced with shPKM2#2. I would have expected a background staining of PKM2, at least in the shPKM2#2-transduced cells. Please confirm that the same laser intensity and detector

sensitivity was used for all the PKM2-stained panels in Figure 3h, and if this is the case, comment on the lack of residual PKM2 staining in shPKM2#2-transduced cells, in contrast with results in Figure S1h.

- In Figure 4, authors claim that “PKM2 aggregation led to the decrease of PKM2 enzymatic activity and disturbances in glycolytic flux”. However, the presented results do not prove that PKM2 aggregation causes the decrease in PKM2 activity, only that both phenomena happen at the same time. Please adjust the conclusions driven from this figure.

- Figure 5: explain the nature of the in-house compound library, its origins, purity, the rationale of choosing these compounds and not others.

- Figure 5a: authors claim that the PKM2 agonists and antagonists can be grouped in six categories, that are not clear. Please, show these categories and explain them.

- Figure 3e and 3g: I understand that the increasing concentrations are referred to K35; but this is not clear in the figure. Please clarify.

- Figure 5g: I can see a clear reduction in p21 protein in cells treated with almost all compounds used: K20, K24, K25, K27, K28, K29 and K34. In many of these, only high concentrations (50 and 100uM) were used, and even with this narrow window of concentrations, the decrease in p21 protein is apparent. Please, clarify why K35 was chosen, and use the same range of concentrations for all compounds in this WB experiment.

- Authors claim that compounds other than K35 are not efficient in alleviating PKM2 aggregation referring to Figure S5d, but in this figure there is only one compounds shown, K1. Please complete this set of experiments with all other compounds.

- Compounds different than K27 are as good as K27 in inhibiting senescence

- Cell lines used in Figure 5 and S5 are very heterogeneous: for IF, HeLa cells are used (Figure 5b); for WB and mRNA, HEK293, MCF7 or fibroblasts (but not HeLa cells). This makes these figures very difficult to follow. Please indicate in each figure panel what precise cell type is being used.

- Figure 5p, r: treatment of young 2BS cells with K35 and K27 increase lactate production and PKM2 activity. However, in principle young 2BS cells do not accumulate PKM2 and,

therefore, this increase in lactate production may indicate an aggregate-independent activation of PKM2 activity by K35 and K27. Please, treat HeLa and HEK293 cells with K35 and K27 but without etoposide, to see if PKM2 activation by these compounds is also seen in non-senescent cells with no PKM2 aggregation.

- Figure S6i: cell viability was measured in young fibroblasts, but not in old fibroblasts where senescence is present and the expected effects of K35 and K27 are stronger. Please, perform viability assays in senescent fibroblasts and in Etoposide-treated HEK293 and HeLa cells treated with K35 and K27.

- Figure S6j-l: these docking analysis suggest that K35 and K27 may bind to PKM2 (even though direct evidence is lacking); but they do not support the notion that they disrupt PKM2 aggregates, as suggested by the authors. Please, explain better or re-phrase the conclusion from these predictions.

- Figure S6m: please, explain why this MST analysis indicates that K35 and K27 bind directly to PKM2 and disrupt its aggregation.

- Figure 6: the doxorubicin treatment is a model of tissue stress due to cytotoxic damage, but in my opinion it is not an aging model. Please comment.

- Figure S7: please indicate the safety and hematological parameters also for K27-treated mice.

- Figure S7c: please indicate how was food intake measured, and why there is no error bar. This information should at least be included in the Methods section.

- Figure S7d: K35 induces a clear decrease in HB at the 50 and 150mg/kg doses. Please comment.

- Figure 6c: rotarod and grip strength are performed 20 days after the first doxorubicin inoculation. However, at this timepoint almost all doxorubicin-treated mice are already dead (Figure 6b). Please, clarify.

- Figure S7g-h are referred to after Figure S8a-h in the text, which is confusing.

- Figure S7g: please underline the senescence-associated genes in the list of differentially expressed genes on this panel. Many of the presented genes are not related to senescence, or their association is not clear to the reader.
- Figure S8e-h: the staining of SA-b-Gal is not apparent in the presented sections. Please use clearer examples of the very dramatic quantifications presented in panels f and h.
- Figure 6j: please indicate in the figure if the effects of K35 and K27 are significant, compared with vehicle-treated mice; and the effect in total lifespan. Also, include a new figure after revision where mortality is higher in all curves.
- Figure 6m is named in the text before Figure 6k and l. Please revise.
- In Discussion, authors write that “the existing form of PKM2, not the amount of the protein, correlates with cellular senescence and aging”. I think authors mean “the active form of PKM2,...”; otherwise, this phrase is not clear to me.

Minor comments:

- In “Western blot”, Western is written with capital “W”.
- In general, gene names should be written in italics (Figure S1c, S2b, Table S1, Figure S2k and other panels/tables).
- Figure S2d: please indicate what CPT stands for (camptothecine, maybe?)

As a summary, the work by Mie and colleagues is interesting for the general public and those specialized in aging, metabolism and senescence. However, there are several points that are not properly explained or interpreted and need to be addressed before publication.

Response to Reviewers' comments:

Reviewer #1 (Remarks to the Author):

The authors reported that PKM2 aggregation drives cell senescence and mice aging. Meanwhile the authors identified two activators which can alleviate PKM2 aggregation and delay aging-related phenotype. The effect of two activators on delaying the aging phenotype is quite significant and can be used as potential anti-aging drugs. However, it lacks sufficient direct evidence to prove the effect of PKM2 aggregation on cell senescence. It is confusing whether two activators influence the aggregation or enzyme activity directly. And it lacks connection between mechanism and phenotype. Overall, the materials and techniques here compromise the novelty. And there are some confusing points need to be explained by the authors.

We really appreciate the positive view to our manuscript by this reviewer to find our work “The effect of two activators on delaying the aging phenotype is quite significant and can be used as potential anti-aging drugs”. The following are the responses to reviewer’s concerns:

1). it lacks sufficient direct evidence to prove the effect of PKM2 aggregation on cell senescence.

Response: We thank the reviewer for raising this point. To address this concern, we looked at our data carefully and also performed several experiments for more convincing evidence to prove the effect of PKM2 aggregation on cell senescence. First, we found that PKM2 truncates with A1, A2 or C domain deletion tended to form aggregates *in vivo* and *in vitro* (Figs. 2f-h), and transfection of these aggregate-prone truncates led to cellular senescence, indicated by higher p21 expression level, higher SASP mRNA level and stronger SA- β -gal staining (Supplementary Figs. 3f-i). Second, we performed new experiments by overexpressing wild-type PKM2 and aggregate-prone PKM2 mutant (A2 domain deletion) in mice via AAV9 system. After 14 weeks, the mice were subjected to physical test and then sacrificed for an evaluation of senescent markers. The results showed that the mice transduced with aggregate-prone PKM2 mutant exhibited weaker physical fitness, upregulated levels of SASP factors and p21, p16 mRNA, which indicated higher degree of aging. The new data are shown in Figs. 2i, 2j and Supplementary Figs. 3j, 3k. All above data directly demonstrated that PKM2 aggregation directly induced cellular senescence or aging *in vivo*. Besides, through two step-screen, we found that K27 and K35 could dissolve the PKM2 aggregates and impede cellular senescence in various cell types (Figs. 5b-n and Supplementary Figs. 6c-l, 7a-c) and alleviate aging phenotypes of mice (Fig. 6 and Supplementary Figs. 8h, 9, 10c-h), which further demonstrated that the PKM2 aggregation is the driving force to inducing cellular senescence or aging *in vivo*.

2). It is confusing whether two activators influence the aggregation or enzyme activity directly.

Response: We thank the reviewer for raising this point. To address this concern, we performed new experiment by treating PKM2 aggregate-free HeLa or HEK 293T cells with K27 and the measured PKM2 enzymatic activities. The new data showed that K27 could not upregulate PKM2 enzymatic activities in young HeLa or HEK 293T cells (Supplementary

Fig. 7i). However, K27 could rescue the downregulated PKM2 enzymatic activities due to PKM2 aggregation (Supplementary Figs. 7f, g). Overall, these results indicated that K27 activated PKM2 by dispersing PKM2 aggregation.

3. it lacks connection between mechanism and phenotype.

Response: We thank the reviewer for raising this issue. However, we think that the connection between mechanism (PKM2 aggregation decreases PKM2 enzymatic activity and disturbances of glycolytic flux) and phenotype (cellular senescence and aging) is clear in this story. The detailed explanations are below:

- 1) We found the phenotypes in which PKM2 forms aggregates in different kinds of senescent cells including: etoposide-induced senescent cells as well as other genotoxic agents-induced senescent cells or ROS-induced senescent cells, and replicative senescent cells, or oncogene-induced senescent human fibroblasts. (Figs. 2a, b and Supplementary Figs. 2d-f) and in aged liver and brain tissues (Fig. 2c and Supplementary Fig. 3a).
- 2) We further found that PKM2 aggregation directly induces cellular senescence or aging *in vivo* (Fig. 2i, j and Supplementary Fig. 3f-k).
- 3) Many studies indicated that glycolytic flux was disturbed during the onset or process of senescence, and senescent phenotypes could be mitigated by restoring glycolysis to normal levels in some way (Kondoh et al., 2005; Kondoh et al., 2007; Minhas et al., 2021). In our study, we found that PKM2 aggregation accompanied the impairment of PKM2 enzymatic activity and glycolytic flux (Fig. 4 and Supplementary Fig. 5), which led to cellular senescence.
- 4) Finally, we screened out the two antagonists of PKM2 aggregation, K35 and K27, which could not only rescue the downregulated PKM2 enzymatic activities and glycolytic flux (Fig. 5o-r and Supplementary Fig. 7d-g) but also impede cellular senescence (Fig. 5c-n and Supplementary Fig. 6c-l, 7a-c) and alleviate aging phenotypes of mice Fig. 6 and Supplementary Fig. 8h, 9, 10c-h) by dissolving the PKM2 aggregation.

All these results demonstrated that PKM2 functions in cellular senescence and aging process by forming aggregation and this aggregation caused downregulation of PKM2 enzymatic activities and glycolytic flux.

4. the materials and techniques here compromise the novelty.

Response: We are sorry that we do not catch on to this statement because that the reviewer did not provide any detail on what materials or techniques compromise the novelty.

Major points:

1. Why does the author use cervical cancer cells HeLa and HEK293T for mechanism research at the cellular level, while focusing on the liver, lungs, and muscles at the individual level? What is the connection between these indicators?

Response: We thank the reviewer for raising this issue. HeLa and HEK293T cells are frequently used as model cell lines in the study of cellular senescence (Wang et al., 2019). Besides the two cell lines, we also repeated the experiments in human fibroblasts (2BS or IMR-90). In the animal models, we chose liver, lung, or kidney organs to reflect the aging status of mice according to previous researches (Bellanti and Vendemiale, 2021; Cai et al., 2020; Schneider et al., 2021; Takemon et al., 2021). Using these cell line and animal models, we not only connected the *in vitro* and *in vivo* models (lung originated 2BS and IMR-90 cells and lung; kidney originated HEK293T cell and kidney), but also demonstrated our results in other cells and organs.

2. Whether PKM2 aggregation only occur during DNA damage-induced senescence? Would PKM2 aggregation occur during ROS-induced or oncogene-induced senescence?

Response: We thank the reviewer for raising this excellent point. Following the suggestion, we performed experiments and demonstrated that PKM2 aggregates also occur during ROS-induced or oncogene-induced senescence. The new data are shown in new Supplementary Figs. 2e, f. Therefore, PKM2 aggregation occurs not only in DNA damage-induced senescence and replicative senescence (Figs. 2a, b and Supplementary Fig. 2d), but also in ROS-induced or oncogene-induced senescence (Supplementary Figs. 2e, f).

3. At the animal level, why does the author not use ETO to induce aging as same as at the cellular level? Otherwise, use both ETO and doxorubicin at the cellular and animal level? The induction should be the same.

Response: We thank the reviewer for raising this point. According to previous studies in the field, doxorubicin was usually used to induce aging in mice (Sun et al., 2022; Dang et al., 2019). As for cellular senescence, etoposide is a common inducer (Georget et al., 2023; te Poele et al., 2002). Therefore, we chose doxorubicin to induce aging in mice. Besides the induced aging mouse model, we also demonstrated our results in naturally aged mice.

4. Through lysosome-MS, the authors found PKM2 and GPI were enriched in lysosomes during the etoposide-induced senescent HEK293T. But why the authors aimed at PKM2 rather than GPI? How to distinguish between PKM2 and its isoenzyme PKM1 through MS? What role does PKM1 play in etoposide-induced senescence?

Response: We thank the reviewer for raising this very important point. The reasons we chose PKM2 for the ensuing investigation are: 1) we found that PKM2 aggregating during senescence is more obvious than GPI; 2) PKM2 is one of the rate-limiting enzymes in glycolysis. Therefore, we focus on PKM2 rather than GPI. Through MS, we identified 11 peptides of PKM sequence, among these 11 peptides, 10 of them are shared and one is unique to PKM2. There are no unique peptides belonging to PKM1. Therefore, PKM2 was enriched in lysosomes. The new data is shown in Supplementary Fig. 1f. We also performed florescent staining experiment to examine the PKM1 and found that PKM1 aggregates also appeared in etoposide-induced senescent HEK 293T cells. The new data is shown in Supplementary Fig. 2g. Consequently, PKM1 aggregation might also involve in cellular senescence and

further experiments are required for the regulation of PKM1 aggregates in cellular senescence in another new story.

5. It has reported that some proteins, eg. SIRT1(Xu et al., 2020) and PKM2(Wu et al., 2023) undergo degradation during cell senescence, so why does PKM2 aggregate in lysosomes and not be degraded? We have reported that the expression of PKM2 in etoposide-induced senescent cells was down-regulated (Wu et al., 2023), here Fig1d showed no change, whether it was due to differences in cell type or something else?

Response: We thank the reviewer for raising this issue. The protein level is dynamic regulated depending on the cellular status. The lysosomal function was impaired in senescent cells (Leeman et al., 2018). In our story, we found that PKM2 aggregates were more solid (resistance to hexanediol). Therefore, PKM2 aggregates might be partially resistant to lysosomal degradation. To further verify our results, we repeated these experiments in etoposide/CPT-induced cells. The results are shown below, (Extra Data Figure 1) which also indicated the unchanged protein levels of PKM2 in DNA damage-induced senescent cells. As for the difference of our results with the results from cited paper, we speculate that the treating condition (drug concentration and exposure time) and cell types were different, which might account for the difference of the results.

Extra Data Figure 1

a. MCF-7 cells were treated with etoposide (ETO, 10 μ M) for indicated days to induce senescence and then subjected to Western blot analysis. **b.** HEK 293T cells were treated with camptothecin (CPT, 25 nM) for 24 hours and released for indicated days to induce senescence and then subjected to western blot analysis. **c.** HeLa cells were treated with etoposide (ETO, 10 μ M) for 24 hours and released for indicated days to induce senescence and then subjected to western blot analysis.

6. In fig1e, the co-location between PKM2 and lysosome is not very well, so Why? It seems that the level of PKM2 in cytoplasm is increased during ETO-induced HEK-293T, nuclear cytoplasmic separation assays should be applied. I doubt that the quantitative method shown by Fig1e-f can be replaced with the area where PKM2 overlaps lysosome vs. the total area occupied by PKM2.

Response: Thanks for the comment. In fig.1e, the reason that the co-location between PKM2 and lysosome is not very well is because not all PKM2 aggregates entered the lysosomes. PKM2 aggregates located at both lysosome and cytoplasm. In the confocal image, the green seems brighter which might be due to the aggregating of PKM2 instead of the increased protein level. We appreciate the reviewer's suggestion for the quantitative method and we did try to use this method to quantify the area where PKM2 overlaps lysosome vs. the total area occupied by PKM2. However, we found that it is hard to achieve because:1) it is very difficult to quantify area in cells; 2) the areas of lysosome signals are varied in different cells and different conditions. Therefore, we chose the quantitative method shown in Fig.1e-f to count the percentage of cells whose lysosomes merged with PKM2 aggregates.

7. Fig2i-j images are not clear, please provide high-quality images, and the images are inconsistent with quantitative data.

Response: Thanks for pointing out this problem. Following the suggestion, we repeated the experiments and the new data are shown in Supplementary Figs. 3h, 3i.

8. Fig3h showed that GAPDH showed aggregation in etoposide-induced senescent cells, and studies showed that the loss of PKM2 would induce cellular senescence, so why GAPDH in PKM2-knockdown cells showed the same distribution state as the control group without etoposide, please explain. In Fig3h it seems that the aggregation of GAPDH depends on PKM2, so does the aggregation of other glycolytic metabolic enzymes (ENO1, GPI, LDH or PFKP) also depend on PKM2? Why can PKM2 drive the aggregation of other metabolic enzymes?

Response: We thank the reviewer for raising this interesting issue. The possible explanation for why GAPDH in PKM2-knockdown cells showed the same distribution state as the control group without etoposide treatment and why can PKM2 drive the aggregation of other metabolic enzymes is as following: 1) According to our MS results of PKM2 aggregates, GAPDH is one of the major components of the PKM2 aggregates (Fig. 3c); 2) PKM2 had been reported to interact with many glycolytic enzymes (Menard et al., 2014). The interaction could account for its ability to drive the aggregation of other metabolic enzymes (Fig. 3d); 3) We demonstrated that PKM2 could aggregate in vitro alone (Figs. 2d, 2h, Supplementary Figs. S3b, S3d), which indicated that PKM2 could be the drive force to form aggregate in cells; 4) We demonstrated that GAPDH aggregates depends on the existence of PKM2 (Fig. 3h). From above four points, we can conclude although loss of PKM2 can induce cellular senescence, GAPDH cannot form aggregates without PKM2. This is the reason why GAPDH in PKM2-knockdown cells showed the same distribution state as the control group without etoposide treatment and PKM2 can drive the aggregation of other metabolic enzymes.

9. The expression of PKM2 shown by Fig3f is down regulated in etoposide-induced senescent cells, which is clearly different from the results in Fig1d and Fig3h, and the authors should give an explain. Why is the PKM2 aggregation state shown by Fig2e and Fig3h so different?

Response: We thank the reviewer for raising this issue and we are sorry for causing the confusing. The expression level of PKM2 shown in Fig. 3f was not downregulated in

etoposide-induced senescent cells, instead, it is due to inconsistency of processing parameters like the output intensity. We re-processed the images and the new data is shown in Fig. 3f which showed the similar PKM2 intensity in Fig. 1d and Fig. 3h. As for the PKM2 aggregation state shown by Fig. 2e and Fig. 3h, this is because that PKM2 aggregation status varies depending on the senescent stages. At the early stage, PKM2 aggregates are rounder and in the later stage, PKM2 aggregates are more amorphous.

10. In Fig5g, why the p21 is higher in ETO Release Day3+10mM K24 than in ETO Release Day3?

Response: We thank the reviewer for raising this issue. The reason might be the amount of β -actin in ETO Release Day 3+10 mM K24 is more than in ETO Release Day 3. We calculated the p21/ β -actin ratio of the two groups by ImageJ, which were 0.46 and 0.41 respectively, which might be more reasonable to reflect efficacy of K24 on p21 level. To avoid this confusing, we repeated the experiment and the new data with more even β -actin level is shown in new Fig. 5g.

11. The growth advantages of HeLa cells showed by the treatment of K35 and K27 in Fig5n-o are inconsistent with the lactate content in the medium.

Response: We thank the reviewer for raising this issue. Fig. 5m, n represent the ability of cell proliferation, while Fig. 5o represent the normalized activities of glycolysis. The two assays reflected different aspects of cells. Consequently, it is not strange that the effects of K35 and K27 were not totally consistent in the two different experiments. However, K35 and K27 improved both cell proliferation and activities of glycolysis are consistent.

12. Supplementary Fig.S1h, The knockdown efficiency of PKM2 is inconsistent with the previous immunofluorescence, and the knockdown efficiency of #2 is not good, resulting in the aging phenotype is more pronounced than #1 in Supplementary Fig.S4l, why?

Response: We thank the reviewer for raising this issue. The cells used in Supplementary Fig.S1h are in the early stage of making knockdown stable cell line and had not been stably selected, while the cells used in Fig. 3h was stable cell lines. Consequently, the knockdown efficiency of PKM2 in Fig. 3h was better than Fig.S1h. We repeated the experiment in Fig.S1h with the same cell line in Fig. 3h. The new data is shown in Supplementary Fig. 1k. As for Supplementary Fig. S4l, the figure was not about PKM2 knock-down, it is the experiment about measure of PKM2 enzymatic activity.

13. Supplementary Fig. S4l showed CPT-induced aging with a decrease in lactate levels in HEK 293T medium is not significantly compared to etoposide.

Response: We thank the reviewer for raising this issue. We repeated the experiment and treated HEK 293T cells with CPT and released for different days followed by evaluation of lactate content in medium. The result revealed that CPT could gradually downregulate lactate production with the extension of CPT-release days. The new data are shown in Supplementary Fig. 5l.

14. In line 871, "ETDA" should be corrected to "EDTA".

Response: We thank the reviewer for pointing out this mistake. We corrected it in the revised manuscript.

Reviewer #2 (Remarks to the Author):

In this work, Bie and colleagues perform lysosome purification followed by proteomics analysis in senescent cells, and identified many glycolytic proteins in senescent cells lysosomes. In particular, they focus on the formation of aggregates of pyruvate kinase M2 (PKM2) that also carry other glycolytic proteins, and are associated to decreased PKM2 activity and glycolytic flux. Authors then perform a drug screening identifying two related compounds (K35 and K27) that disperse PKM2 aggregates and decrease senescent markers in vitro. They then test these compounds in a model of chemotherapy stress and in naturally aged mice, and again show decreased senescence markers and improved survival and neuromuscular performance in K35- and K27-treated mice.

In general, the work is interesting and novel, showing a possible new mechanism of metabolic evolution of cells as they age/senesce that can help explaining the metabolic shift in this physiological process. There are however several points unclear in the description of the results and their interpretation, specially in the last part of the manuscript, that must be clarified before publication.

We really appreciate the positive view to our manuscript by this reviewer to find our work is “interesting and novel, showing a possible new mechanism of metabolic evolution of cells as they age/senesce that can help explaining the metabolic shift in this physiological process”. The following are the responses to reviewer’s concerns:

General remarks:

- Letter size in many figure panels is too small and hard to read.

Response: We apologize for the inconvenience to the reader. We have adjusted the letter size in those figure panels in the revised manuscript.

- I cannot see a proper Statistical analysis report in the Methods section. In particular, I can see in the Figure legends that two-tailed Student t-test was performed in several figures where several groups are compared (Figure 1f, 2j, 4h-m, 5d-r, 5c-p), and a one-way ANOVA should have been used; and no statistical analysis method is indicated for Figures 6a (two-way ANOVA would fit) and 6b (logrank test). Please re-do these analyses and use an appropriate test for each panel, indicating it in the Figure legends and in a complete Statistical analysis sub-section in the Methods section.

Response: We thank the reviewer for this great suggestion. In the revised manuscript, we re-did these analyses using an appropriate test for each panel, which was indicating in the Figure legends and we also added a complete Statistical analysis sub-section in the Methods section.

- Authors claim that compounds K35 and K27 reduce senescence markers in many cells, including p21 and/or p16 protein, SASP-associated cytokine transcription and SA-b-Gal. Is this decrease due to

senescence reversal, or due to senescence cell death (senolytic effects)? In principle, senescence is considered a very stable cellular status (irreversible for many), and this point is rather controversial. Comparison of cell viability between proliferating and senescent cells treated with vehicle or K35/K27 but without chemotherapy administration would clarify this point.

Response: We thank the reviewer for raising this important point. Indeed, cell viability was measured in young and old fibroblasts in Supplementary Fig. 6i (new Supplementary Fig.7j). And we also performed viability assays in DMSO or etoposide-treated HeLa and HEK 293T cells treated with K35 and K27. The new data are shown in Supplementary Fig. 7k, l. All these data showed that both young or senescent cells treated with K35 or K27 were in the same trends, which didn't support senolytic effects of K35 or K27. As for the term "reverse aging", in recent years, many scientists started to declare that senescence or aging could be reversed (Adler et al., 2008; Walters et al., 2016; Yang et al., 2023).

- The Methods section is confusing, alternating in vitro, in vivo and in silico data. Please, group the methods to facilitate reading.

Response: We thank the reviewer for this helpful suggestion. We rearranged the method section in the revised manuscript according to the suggestion.

- No effort is made to evaluate pharmacological parameters of K35 and K27. Bioavailability and tissue distribution of these compounds should be performed, including their ability to cross the blood-brain barrier. This is specially important since these compounds are administered intragastrically, and it is not certain if they resist the digestive tract or are absorbed into the bloodstream.

Response: We thank the reviewer for raising this important issue. Following the suggestion, we evaluated the concentration of K35 and K27 in serum and brain of mice by quantitative LC-MS assay. The new data are shown in Supplementary Figs. 8b, c. These results indicated that K35 and K27 resisted the digestive tract and were absorbed into the bloodstream. Also, K35 and K27 had the ability to cross the blood-brain and distributed in the brain of mice.

Major Comments:

- Figure S1: there is HA-tagged TMEM192 in the ER compartment, but no mention is done in the text or the figure legends of this apparent impurity of the IP. Please comment.

Response: We thank the reviewer for the helpful comments. ER was reported to interact with lysosome (D. Höglinger et al., 2019). It is normal to find ER marker in the IP sample. Such "impurity" (CALR band) could also be found in immunoprecipitated lysosomes of other researches utilizing this method (Kumar et al., 2022; Liu et al., 2023).

- In the Methods section for lysosome IP, authors indicate that HeLa cells stably expressed TMEM192-HA; but in the figures, authors indicated 293T cells. Please verify.

Response: We thank the reviewer for point out this mistake. HeLa should be HEK 293T. We corrected it in the revised manuscript.

- Figure 1d: it seems that the b-actin loading control of the input blots to the right increased with time of treatment. Please quantify the increase in the HA-IP proteins relative to b-actin loading control, using replicates for each time point.

Response: We thank the reviewer for raising this issue. We repeated this experiment using replicates for each time point. In HA-IP samples, LAMP2 should be the loading control. Consequently, we quantified the increase of those proteins in immunoprecipitated lysosomes relative to LAMP2 loading control and the new data is shown in new Fig. 1d.

- Figure 1e, f and S1f: please indicate what dye/marker was used for the identification of lysosomes, and how was the quantification of the merge between PKM2/GPI and lysosomes performed. I cannot see this information in the Methods section. Also, quantify the colocalized GPI-lysosomes for Figure S1f, as shown for PKM2 in Figure 1f.

Response: We thank the reviewer for the helpful suggestions. We indicated the dye/marker used for the identification of lysosomes in figure legends. As for the quantification, we clarified the quantification details in the method section. We also did the same quantification for Figure S1f (Supplementary Fig. 1g in revised version) and the new data is shown in Supplementary Fig. 1h.

- Figure S1g-h: please, indicate the exact sequences of the shPKM2 used in these figures.

Also, please quantify, with an appropriate number of replicates (≥ 3), the alleged increase in SA-bGal (Figure S1g) and of p21 (figure S1h) in the shPKM2-infected cells. In particular, shown images of the SA-bGal staining are not very clear: very few cells are shown for the DMSO baseline controls for the shPKM2-infected cells, showing already high SA-bGal staining. Quantification of large number of cells, relative to SA-bGal-positive cells, should be provided to prove an increase in senescence. Also, since GAPDH is altered in senescence, I would recommend using a different loading control for WB in Figure S1h, such as b-actin, as in Figure 1d.

Response: We thank the reviewer for the helpful suggestions. Indeed, we did indicate the exact sequence of shPKM2 used in the supplementary materials. We also repeated all the indicated experiments and quantified the p21 protein levels and SA- β Gal staining in Figure S1g and figure S1h. We change the loading control from GAPDH to β -actin in Figure S1h. The new data is shown in Supplementary Figs. 1i-k.

- Figure S2f: aggregation of sfcherry-PKM2 is not very clear in these images, with some aggregation (stronger staining points) also visible at T0 in sfcherry-PKM2. Please provide clearer images, or quantify the aggregation.

Response: We thank the reviewer for the helpful suggestion. We quantified the aggregation in the revised manuscript. The new data is shown in Supplementary Fig. 2i.

- Figure 2d and S2g: please clarify which cells were used to overexpress sfcherry-PKM2, and how was the purification performed. In the Methods section, only flow cytometry analysis of sfcherry-PKM2 is indicated. Also, indicate how different salt concentrations were reached and monitored.

Response: We thank the reviewer for the helpful comments. The sfcherry-PKM2 was purified using bacterial system. We have explained this in the method part of the revised manuscript. The salt concentration was determined by mixing sfcherry-PKM2 with salt solution whose salt concentration is clear. We have indicated this in the method part of the revised manuscript.

- Figure 2e: authors claim that “the confocal images (Fig. 1e) showed that not all PKM2 aggregates merged with lysosomes”. However, no lysosome marker is used in these IF images, so this claim is not appropriate. Also, please quantify PKM2 and lysosome or mitochondria co-localization, as done before with Figure 1f, to generate a robust piece of data. Finally, please indicate in the figure legend what does the CS (citrate synthase) marker for mitochondria identification stand for.

Response: We thank the reviewer for the helpful suggestions. In Fig. 1e, since we made the HA-tagged TMEM19-3 stable HEK 293T cell line, we used this cell line to stain HA to represent lysosomes. We indicated this in figure legends of the revised manuscript. From Figure 2e, we can clear see no apparent merge of PKM2 and mitochondria was observed, therefore, we didn’t quantify the co-localization of PKM2 and mitochondria. CS (citrate synthase) used as mitochondria marker and we clarified it in the figure legends of the revised manuscript.

- Figure S2h: please quantify PKM2 and stress granule co-localization, as done before with Figure 1f, to generate a robust piece of data.

Response: We thank the reviewer for the comment. From Figure S2h (Supplementary Fig. 3c in the revised version), we can clear see that under the low concentration of etoposide treatment for inducing senescence, there is no apparent stress granule (stain of G3BP1) induced, therefore, we didn’t quantify the co-localization of PKM2 and stress granule co-localization.

- Figure S2j: please clarify why the different patterns of Coomassie staining of the truncate mutants of PKM2 illustrate differential aggregate formation in vitro. Also, as requested for Figure 2d and S2g, clarify the protocol followed to obtain these cherry-PKM2 aggregates in vitro.

Response: We thank the reviewer for raising this issue. Actually, the Coomassie staining in Figure S2j (Supplementary Fig. 3e in the revised version) is just for indicating the concentration and the quality of PKM2 truncates which were used in aggregation assay in vitro (Figure 2h). Consequently, Figure 2h together with Figure S2j but not Figure S2j alone illustrated differential aggregate formation of truncated PKM2 in vitro. The sfcherry-PKM2 FL and truncate mutants were purified using bacterial system. We have clarified the protocol in the method part of the revised manuscript.

- In Figure 3h, no PKM2 staining is observed in the IF of shPKM2 #1 and #2. However, in Figure S1h, a significant amount of PKM2 was observed by WB, specially in cells transduced with shPKM2#2. I would have expected a background staining of PKM2, at least in the shPKM2#2-transduced cells. Please confirm that the same laser intensity and detector sensitivity was used for all the PKM2-stained panels in Figure 3h, and if this is the case, comment on the lack of residual PKM2 staining in shPKM2#2-transduced cells, in contrast with results in Figure S1h.

Response: We thank the reviewer for raising this issue. The reason for this is due to the cells used in Fig.S1h are in the early stage of making knockdown stable cell line and had not been stably selected, while the cells used in Fig3h was stable cell lines. Consequently, the knockdown efficiency of PKM2 in Fig3h was better than Fig.S1h. We repeated the experiment in Fig. S1h with the same cell line in Fig 3h. The new data is shown in Supplementary Fig. 1k.

- In Figure 4, authors claim that “PKM2 aggregation led to the decrease of PKM2 enzymatic activity and disturbances in glycolytic flux”. However, the presented results do not prove that PKM2 aggregation causes the decrease in PKM2 activity, only that both phenomena happen at the same time. Please adjust the conclusions driven from this figure.

Response: We thank the reviewer for the helpful suggestions. For Figure 4, we have rephrased the subtitle and the conclusion in the revised manuscript to avoid overstatement. Now, the conclusion is “PKM2 enzymatic activity and glycolytic flux are impaired in senescent cells, accompanying with PKM2 aggregation.”

- Figure 5: explain the nature of the in-house compound library, its origins, purity, the rationale of choosing these compounds and not others.

Response: We thank the reviewer for the helpful suggestions. We explained compound library in the method section of the revised manuscript. The compound library we selected comes from the research group of Dr. Ridong Li, the co-first author of this article. His research group has been dedicated to the study of PKM2 agonists and PKM inhibitors for many years. Considering that the research content of this article is related to PKM2, the compounds selected in the first round of screening were all PKM2 agonists or PKM inhibitors. The structures of these compounds were verified by nuclear magnetic resonance hydrogen spectroscopy (¹H NMR) and nuclear magnetic resonance carbon spectroscopy (¹³C NMR), and their purity was greater than 95%.

- Figure 5a: authors claim that the PKM2 agonists and antagonists can be grouped in six categories, that are not clear. Please, show these categories and explain them.

Response: We thank the reviewer for the helpful suggestions. We showed the six categories in supplementary fig. S6b and explain them in the revised manuscript.

- Figure 3e and 3g: I understand that the increasing concentrations are referred to K35; but this is not clear in the figure. Please clarify.

Response: We thank the reviewer for the helpful suggestions. The reviewer should refer to Figure 5e and 5g instead of 3e and 3g. We adjusted the figure panels to make it clearer. The new data are shown in Figs. 5e, 5g.

- Figure 5g: I can see a clear reduction in p21 protein in cells treated with almost all compounds used: K20, K24, K25, K27, K28, K29 and K34. In many of these, only high concentrations (50 and 100uM) were used, and even with this narrow window of concentrations, the decrease in p21 protein is apparent. Please, clarify why K35 was chosen, and use the same range of concentrations for all compounds in this WB experiment.

Response: We thank the reviewer for the helpful suggestions. To identify the compounds which could disperse PKM2 aggregates and alleviate senescence, we designed a two-step screening program to search for the ideal compounds. First, an in-house small molecule library was applied to identify PKM2 agonists and antagonists which had higher possibilities to lessen PKM2 aggregation. These agonists and antagonists could be divided into six categories according to their structures. Second, the six representative compounds from each category were selected to evaluate their effects on PKM2 aggregation and senescence-associated phenotypes. One of the six compounds, K35, was found to be the desired compound in dissipating PKM2 aggregates and alleviating senescence-associated phenotypes while the other five types of PKM2 activators or inhibitors had little effect on lessening PKM2 aggregation or even aggravating it (Supplementary Fig. S6e). For more potent compounds, eight analogs of K35 including K19, K20, K24, K25, K27, K28, K29 and K34 were evaluated for influence on PKM2 aggregation and senescent phenotypes which led to the identification of K27 as the most efficient compound. Consequently, we picked K35 and K27 for further investigation.

Because we have found 50 or 100 μM of K35 have significant effects on cellular senescence (Figure 5e). Therefore, we chose 50 and 100 μM for most compounds to evaluate their effects on p21 protein level. We repeated the experiments using the same range of concentrations for all compounds. The new data is shown in Fig. 5g.

- Authors claim that compounds other than K35 are not efficient in alleviating PKM2 aggregation referring to Figure S5d, but in this figure there is only one compounds shown, K1. Please complete this set of experiments with all other compounds.

Response: We thank the reviewer for the helpful suggestions. Actually, we already performed the experiment with all other compounds from six categories identified. The new data is shown in Supplementary Fig. 6e in the revised manuscript.

- Compounds different than K27 are as good as K27 in inhibiting senescence

Response: We thank the reviewer for raising the point. We admit that other analogs also performed well in inhibiting cellular senescence. But according results depicted in the revised Fig. 5g and revised Supplementary Fig. 6f-6j, K27 exhibited the most powerful ability in inhibiting cellular senescence. Consequently, we chose K27 for further analysis.

- Cell lines used in Figure 5 and S5 are very heterogeneous: for IF, HeLa cells are used (Figure 5b); for WB and mRNA, HEK293, MCF7 or fibroblasts (but not HeLa cells). This makes these figures very difficult to follow. Please indicate in each figure panel what precise cell type is being used.

Response: We thank the reviewer for pointing out this problem. We have indicated in each figure panel what precise cell type was being used in the revised manuscript.

- Figure 5p, r: treatment of young 2BS cells with K35 and K27 increase lactate production and PKM2 activity. However, in principle young 2BS cells do not accumulate PKM2 and, therefore, this increase in lactate production may indicate an aggregate-independent activation of PKM2 activity by K35 and K27. Please,

treat HeLa and HEK293 cells with K35 and K27 but without etoposide, to see if PKM2 activation by these compounds is also seen in non-senescent cells with no PKM2 aggregation.

Response: We thank the reviewer for the helpful suggestions. In relatively young 2BS cells, there are fewer but not no PKM2 aggregates. Consequently, treatment of young 2BS cells with K35 and K27 slightly increase lactate production and PKM2 activity. As you mentioned, we have treated HeLa and HEK 293T cells with K35 and K27 but without etoposide, to see if PKM2 activation by these compounds is also seen in non-senescent cells with no PKM2 aggregation. The enzymatic assay indicated that only high concentration of K35 could slightly activate PKM2 in non-senescent HeLa or HEK 293T cells, while K27 could not. Consequently, K35 might own an aggregate-independent activation of PKM2 activity, while K27 might not. The new data is shown in revised Supplementary Fig. 7i.

- Figure S6i: cell viability was measured in young fibroblasts, but not in old fibroblasts where senescence is present and the expected effects of K35 and K27 are stronger. Please, perform viability assays in senescent fibroblasts and in Etoposide-treated HEK293 and HeLa cells treated with K35 and K27.

Response: We thank the reviewer for these helpful suggestions. Actually, cell viability was also measured in old fibroblasts in Figure S6i (Supplementary Fig. 7j in the revised version). And we have performed viability assays in DMSO or etoposide-treated HEK 293T and HeLa cells treated with K35 and K27. The new data are shown in Supplementary Figs. 7k-l.

- Figure S6j-l: these docking analysis suggest that K35 and K27 may bind to PKM2 (even though direct evidence is lacking); but they do not support the notion that they disrupt PKM2 aggregates, as suggested by the authors. Please, explain better or re-phrase the conclusion from these predictions.

Response: We thank the reviewer for the helpful comments. In Figure S6m (Supplementary Fig. 7p in the revised version), we showed K35 and K27 directly bind to PKM2 via MST assays, which is the direct evidence. According to the docking results, the two compounds localize at the interface of two PKM2 monomer. Therefore, we speculated that they might disrupt PKM2 aggregates binding and remolding the PKM2 structure according to these results. We have explained better and more in the revised manuscript.

- Figure S6m: please, explain why this MST analysis indicates that K35 and K27 bind directly to PKM2 and disrupt its aggregation.

Response: We thank the reviewer for raising the point. Microscale thermophoresis (MST) is a biophysical assay to quantify the direct interaction between molecules, such as proteins and small molecules (Huang and Zhang, 2021). The MST analysis of K35/K27 with PKM2 demonstrated that PKM2 directly bind to the two compounds. According to the docking results and MST assays, the two compounds localize at the interface of two PKM2 monomers to prevent them forming dimer. Therefore, we speculated that they might disrupt PKM2 aggregates binding and remolding the PKM2 structure according to these results. We have explained better and more in the revised manuscript.

- Figure 6: the doxorubicin treatment is a model of tissue stress due to cytotoxic damage, but in my opinion it is not an aging model. Please comment.

Response: We thank the reviewer for raising this point. According to previous studies, doxorubicin was usually used to induce aging in mice (Sun et al., 2022; Dang et al., 2019). Consequently, we choose the same treatment condition (low dosage of doxorubicin) to establish the accelerating aging model.

- Figure S7: please indicate the safety and hematological parameters also for K27-treated mice.

Response: We thank the reviewer for the helpful suggestions. We performed the safety assay, routine blood tests and serum biochemical examinations for K27 or K35-treated mice at the same time. The new data are shown in revised Supplementary Figs. 8d-g.

- Figure S7c: please indicate how was food intake measured, and why there is no error bar. This information should at least be included in the Methods section.

Response: We thank the reviewer for the comments and helpful suggestions. We summarized the food or water intake of each mouse in the same group, so there was no error bar. We have repeated the assay and the food or water intake of each mouse was counted. The new data are shown in revised Supplementary Fig. 8e. We also clarified how food or water intake was measured in the Methods section of the revised manuscript.

- Figure S7d: K35 induces a clear decrease in HB at the 50 and 150mg/kg doses. Please comment.

Response: We thank the reviewer for raising this point. To verify it, we repeated the assay in K35 or K27-treated mice and monitored the whole process carefully. The result showed that there was no significant difference between vehicle and K35/K27 treated group. The new data is shown in revised Supplementary Fig. 8f. We think there might be something wrong in HB measurement for the first time.

- Figure 6c: rotarod and grip strength are performed 20 days after the first doxorubicin inoculation. However, at this timepoint almost all doxorubicin-treated mice are already dead (Figure 6b). Please, clarify.

Response: We thank the reviewer for raising this point. The mice used in the two assays are not from the same one experiment. The survival of doxorubicin-treated mice was a little different in every different run of the experiment. When we performed the rotarod and grip strength tests, not all mice died before 20 days. And, we picked the living mice to finish the experiments.

- Figure S7g-h are referred to after Figure S8a-h in the text, which is confusing.

Response: We thank the reviewer for raising this issue. This is due to the difficulty of arrangement of figures. To avoid the confusing, we rearranged the figures and correct the figure order in the relevant texts.

- Figure S7g: please underline the senescence-associated genes in the list of differentially expressed genes on this panel. Many of the presented genes are not related to senescence, or their association is not clear to the reader.

Response: We thank the reviewer for raising this issue. All these genes in the figure are related to senescence or aging according to previous studies with KEGG or GO annotation.

- Figure S8e-h: the staining of SA-b-Gal is not apparent in the presented sections. Please use clearer examples of the very dramatic quantifications presented in panels f and h.

Response: We thank the reviewer for the helpful suggestions. We replaced them with more clear images. The new data are shown in revised Supplementary Figs. 9e, 9g.

- Figure 6j: please indicate in the figure if the effects of K35 and K27 are significant, compared with vehicle-treated mice; and the effect in total lifespan. Also, include a new figure after revision where mortality is higher in all curves.

Response: We thank the reviewer for this great suggestion. We performed statistical analysis to indicate the significance and clarified the effects of K35 and K27 in total lifespan. And we also updated this figure in the revised manuscript. The new data is shown in revised Fig. 6j.

- Figure 6m is named in the text before Figure 6k and l. Please revise.

Response: We thank the reviewer for the suggestion. This is due to the difficulty of arrangement of figures. We rearranged the figures and changed the letter order in the text of revised manuscript.

- In Discussion, authors write that “the existing form of PKM2, not the amount of the protein, correlates with cellular senescence and aging”. I think authors mean “the active form of PKM2,...”; otherwise, this phrase is not clear to me.

Response: We thank the reviewer for the helpful comments. You are right. We did mean the active form of PKM2. We replaced the word “existing” with “active” in the revised manuscript.

Minor comments:

- In “Western blot”, Western is written with capital “W”.

- In general, gene names should be written in italics (Figure S1c, S2b, Table S1, Figure S2k and other panels/tables).

- Figure S2d: please indicate what CPT stands for (camptothecine, maybe?)

Response: We thank the reviewer for raising these mistakes. We have corrected these in the revised manuscript.

As a summary, the work by Mie and colleagues is interesting for the general public and those specialized in aging, metabolism and senescence. However, there are several points that are not properly explained or interpreted and need to be addressed before publication.

Response: We thank this reviewer for all the insightful suggestions. We think the revised manuscript had addressed all the concerns raised by this reviewer and is suitable for publication.

References

- Adler, A.S., Kawahara, T.L., Segal, E., and Chang, H.Y. (2008). Reversal of aging by NFkappaB blockade. *Cell Cycle* 7, 556-559.
- Bellantini, F., and Vendemiale, G. (2021). The Aging Liver: Redox Biology and Liver Regeneration. *Antioxid Redox Signal* 35, 832-847.
- Cai, Y., Zhou, H., Zhu, Y., Sun, Q., Ji, Y., Xue, A., Wang, Y., Chen, W., Yu, X., Wang, L., et al. (2020). Elimination of senescent cells by beta-galactosidase-targeted prodrug attenuates inflammation and restores physical function in aged mice. *Cell Res* 30, 574-589.
- Georget, M., Defois, A., Guiho, R., Bon, N., Allain, S., Boyer, C., Halgand, B., Waast, D., Grimandi, G., Fouasson-Chailloux, A., et al. (2023). Development of a DNA damage-induced senescence model in osteoarthritic chondrocytes. *Aging (Albany NY)* 15, 8576-8593.
- Huang, L., and Zhang, C. (2021). Microscale Thermophoresis (MST) to Detect the Interaction Between Purified Protein and Small Molecule. *Methods in molecular biology (Clifton, NJ)* 2213, 187-193.
- Kondoh, H., Leonart, M.E., Gil, J., Wang, J., Degan, P., Peters, G., Martinez, D., Carnero, A., and Beach, D. (2005). Glycolytic enzymes can modulate cellular life span. *Cancer research* 65, 177-185.
- Kondoh, H., Leonart, M.E., Nakashima, Y., Yokode, M., Tanaka, M., Bernard, D., Gil, J., and Beach, D. (2007). A High Glycolytic Flux Supports the Proliferative Potential of Murine Embryonic Stem Cells. *Antioxidants & Redox Signaling* 9, 293-299.
- Kumar, G., Chawla, P., Dhiman, N., Chadha, S., Sharma, S., Sethi, K., Sharma, M., and Tuli, A. (2022). RUFY3 links Arl8b and JIP4-Dynein complex to regulate lysosome size and positioning. *Nature communications* 13, 1540.
- Liu, S., Perez, P., Sun, X., Chen, K., Fatirchorani, R., Mammadova, J., and Wang, Z. (2023). MLKL polymerization-induced lysosomal membrane permeabilization promotes necroptosis. *Cell Death Differ.*
- Minhas, P.S., Latif-Hernandez, A., McReynolds, M.R., Durairaj, A.S., Wang, Q., Rubin, A., Joshi, A.U., He, J.Q., Gauba, E., Liu, L., et al. (2021). Restoring metabolism of myeloid cells reverses cognitive decline in ageing. *Nature* 590, 122-128.
- Schneider, J.L., Rowe, J.H., Garcia-de-Alba, C., Kim, C.F., Sharpe, A.H., and Haigis, M.C. (2021). The aging lung: Physiology, disease, and immunity. *Cell* 184, 1990-2019.
- Takemon, Y., Chick, J.M., Gerdes Gyuricza, I., Skelly, D.A., Devuyst, O., Gygi, S.P., Churchill, G.A., and Korstanje, R. (2021). Proteomic and transcriptomic profiling reveal different aspects of aging in the kidney. *Elife* 10.
- te Poele, R.H., Okorokov, A.L., Jardine, L., Cummings, J., and Joel, S.P. (2002). DNA damage is able to induce senescence in tumor cells in vitro and in vivo. *Cancer research* 62, 1876-1883.
- Walters, H.E., Deneka-Hannemann, S., and Cox, L.S. (2016). Reversal of phenotypes of cellular senescence by pan-mTOR inhibition. *Aging (Albany NY)* 8, 231-244.

Wang, Y., Liu, J., Ma, X., Cui, C., Deenik, P.R., Henderson, P.K.P., Sigler, A.L., and Cui, L. (2019). Real-time imaging of senescence in tumors with DNA damage. *Sci Rep* 9, 2102.

Yang, J.H., Petty, C.A., Dixon-McDougall, T., Lopez, M.V., Tyshkovskiy, A., Maybury-Lewis, S., Tian, X., Ibrahim, N., Chen, Z., Griffin, P.T., et al. (2023). Chemically induced reprogramming to reverse cellular aging. *Aging (Albany NY)* 15, 5966-5989.

REVIEWERS' COMMENTS

Reviewer #1 (Remarks to the Author):

Authors have answered all my comments and revised the document.

Reviewer #2 (Remarks to the Author):

Authors answered satisfactorily all my comments. I consider this manuscript suitable for publication in its revised version.